# ONLINE IN-CONTEXT DISTILLATION
# FOR LOW-RESOURCE VISION LANGUAGE MODELS

## ABSTRACT

As the field continues its push for ever more resources, this work turns the spotlight on a critical question: how can vision-language models (VLMs) be adapted to thrive in low-resource, budget-constrained settings? While large VLMs offer strong performance, they are impractical to deploy in such settings. Small VLMs, on the other hand, are efficient but typically require costly fine-tuning to close the performance gap with larger models in the deployment domain. Inspired by the in-context learning framework, we propose an online In-Context Distillation (ICD) method, in which a small VLM collaborates with a stronger teacher model at inference time, distilling its knowledge via sparse demonstrations to efficiently bridge the gap between them. Our method is built on an in-depth analysis that identifies the scale and the choice of models for which vision-language ICL is currently feasible, and demonstrates the advantage of ICL over fine-tuning under constrained compute budgets. We enhance our method with a novel cross-modal demonstration selection strategy, teacher test-time scaling to reduce noise, and student uncertainty conditioning to dynamically populate a demonstration pool and minimize teacher queries. Our ICD method significantly boosts the performance of small models (up to 33%) using scarce teacher annotations (as low as 4%), and competes with the teacher's zero-shot performance.

## 1 INTRODUCTION

Although state-of-the-art vision-language models (VLMs) (Achiam et al., 2023; Team et al., 2023; Li et al., 2024a) continue to scale in size for improved performance, smaller VLMs remain more practical for several real-world applications and users, due to their accessibility and efficiency. In online deployment scenarios with strict constraints on compute, memory, and latency (e.g., edge devices or mobile applications), small models may even represent the only feasible option. However, reducing the model size typically comes at the expense of a drop in performance, leading to a significant gap between small and large models, especially for unseen tasks during pretraining.

Given an online data stream of a specific task, a common strategy to enhance small models is fine-tuning (Zhou et al., 2022; Rasheed et al., 2023), which in general consists of two steps: data annotation and model training. Human annotation is inefficient for real-time applications as it is expensive and time-consuming (Douglas et al., 2023; Ding et al., 2022). On the other hand, fine-tuning a pretrained model for a new task in an online manner is challenging due to not only the additional computational cost, but also the risk of catastrophic forgetting (French, 1999), which degrades the model's original capabilities (Kumar et al., 2022). An alternative is to apply in-context learning (ICL) (Brown et al., 2020; Dong et al., 2022) techniques to enhance the model. ICL leverages demonstrations as context at inference time to guide models to generate task-specific output without parameter update. Nevertheless, demonstrations still require annotations.

We propose a novel online in-context distillation (ICD) framework to bridge the gap between large and small VLMs in an annotation- and training-free manner. As shown in Fig. 1 (left), our ICD consists of two key components: a powerful teacher VLM that generates low-cost, on-the-fly demonstrations without the need for human supervision, and a lightweight student VLM that leverages these demonstrations via in-context learning at inference time. This approach enables the student to distill knowledge from the teacher in an online manner, improving the performance without retraining and thus satisfying the low compute budget requirement, as shown in Fig. 1 (right). Leveraging larger models as an inexpensive instant source of supervision (Wang et al., 2024; Gilardi et al., 2023)

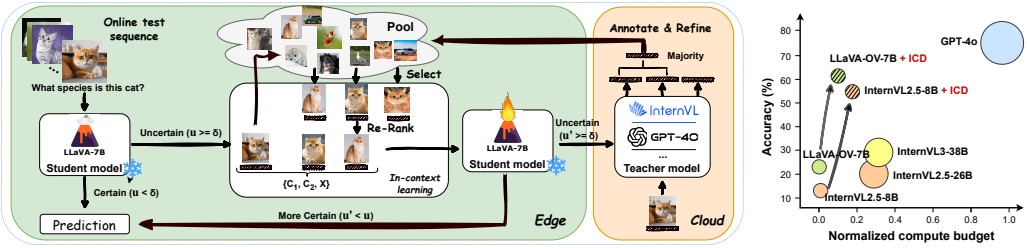

Figure 1: *Left: Overview of our online In-Context Distillation framework. The teacher model in the cloud receives demonstration requests from the student and provides refined answers with test-time scaling. The small student model accesses the dynamically populated pool for answering queries and upon uncertainty, it queries the teacher for new demonstrations. Right: Performance vs. compute budget normalized by that of GPT-4o. Our ICD boosts small VLMs toward the performance of large teacher models with moderate compute overhead.*

is efficient, automatic, and does not bring compute overhead for the student at inference time. We further equip ICD with three modules: (1) test-time scaling techniques (TTS) (Snell et al., 2024) of the teacher to reduce annotation noise and hallucination, (2) cross-modal demonstration retrieval for robust demonstration selection, and (3) uncertainty-based querying that significantly caps annotation budget. Overall, ICD features a modular framework for resource-constrained scenarios of small-large model collaboration, operating on an online pipeline.

A prerequisite for our ICD framework is the effectiveness of ICL in small VLMs. While prior work has demonstrated the success of ICL in large VLMs (Sun et al., 2024; Awadalla et al., 2023) and large language models (LLMs) (Liu et al., 2021; Li et al., 2024c), its utility for small VLMs remains unclear (Zong et al., 2024). To address this gap, we conduct an in-depth analysis of how small VLMs can benefit from ICL. Our findings show that its effectiveness depends heavily on the model's inherent capabilities. Specifically, we find that while models with fewer than 4 billion parameters (denoted as *tiny models*) struggle to benefit from ICL, potentially lacking the fundamental capabilities required for effective ICL, small models (between 4B and 12B parameters) offer a unique advantage of strong ICL capability and a budget-friendly compute. We further trace back ICL failures to a lack of fundamental capabilities such as vision-text alignment, single- and multi-modal reasoning, and visual perception. Interestingly, we find that while the underlying LLM of a VLM can solve certain ICL reasoning tasks, the full VLM fails to do so, suggesting that ICL capabilities in LLMs do not transfer directly to VLMs. We finally show that under a limited compute budget, ICL can bring stronger performance gains compared to finetuning alternatives. These observations serve as the foundation of our efficient and effective approach to boost the performance of a small VLM.

We validate our method with several small VLMs for a range of tasks, including classification, VQA, and image captioning. Our ICD provides a significant performance boost without requiring additional human annotations or model retraining, as well as without frequently querying the heavy-weight teacher. For instance, ICD boosts student performance of 7B model LLaVA-OneVision from 42.6% to 70.8%, compared to zero-shot or TTS-based baselines with as little as 4.4% annotation rate, further outperforming GPT-4o teacher scoring 68.1% Our contributions can be summarized as follows:(1) we conduct an in-depth analysis of ICL capabilities for small VLMs, identifying crucial factors for effective ICL; (2) based on our analysis, we propose a novel online in-context distillation (ICD) framework, designed for real-time compute-constrained scenarios, that combines sparse demonstration querying of a strong model with the inherent ICL capabilities of the student model; and (3) we test our framework across diverse tasks—from classification to generation—demonstrating its broad applicability while boosting base models performance by an average of 14.8% with minimal compute overhead, requiring teacher queries for only 14.7% average of inputs on average.

## 2 RELATED WORK

**In-context learning.** ICL was first proposed for enhancing LLM by providing demonstrations at inference time (Dong et al., 2022), resulting from the emerging properties (Wei et al., 2022a;b). Specifically, it prepends demonstrations to the query sample as illustrative guidance, leading to a significant performance boost in the final generation. Various investigations have been conducted to better understand this mechanism (Dai et al., 2022; Akyürek et al., 2022; Wang et al., 2023b; Pan, 2023) and to enhance this capability (Lu et al., 2021; Liu et al., 2024; Chen et al., 2023a).

**In-context learning for VLMs.** ICL has gained traction in the computer vision community in recent years (Awadalla et al., 2023; Wang et al., 2023a; Chen et al., 2025b) where its potential in

enhancing visual understanding is actively explored. The model leverages image-text pairs as context to infer the desired output for the query. While this ICL capability is believed to be inherited from the LLMs in the VLMs, its effectiveness is less clear in VLMs. It has been shown that additional in-context tuning can further improve the ICL capability (Zhao et al., 2023; Chen et al., 2023b). Other works (Zhang et al., 2023b; Zhou et al., 2024b; Baldassini et al., 2024) investigated the strategies to informative select demonstrations for visual ICL and (Chen et al., 2024) discovered a negative contribution of images in the demonstration for visual ICL. A recent benchmark VL-ICL (Zong et al., 2024) is proposed to better evaluate the ICL capabilities of large VLMs, highlighting the challenges in visual ICL. Nevertheless, the analysis for small VLMs remains underexplored.

**VLMs distillation.** Knowledge distillation (Xu et al., 2024) allows for transferring the knowledge from a large capable model to a smaller efficient model. Such properties are attractive for VLMs where their substantial size and computational demands pose challenges for deployment in resource-constrained environments (Li et al., 2023b; Hu et al., 2024; Wang et al., 2022). Recent works discovered that VLMs models can benefit from the teacher's output, in forms of instructions(Zhou et al., 2024a) or chain of thoughts (Li et al., 2023a). However, existing methods mostly rely on additional training for distillation, which requires annotated data and model training and leads to catastrophic forgetting(French, 1999). A training-free distillation is not yet proposed for VLMs.

## 3  CAN WE APPLY ICL TO SMALL VLMS?

In this section, we analyze the ICL capabilities of various families of small VLMs. We focus on three key questions: (1) which models (and at what scale) can benefit from ICL; (2) using a reasoning task as a use case, what failure modes affect Vision-ICL compared to its pure language counterpart; and (3) whether ICL can be advantageous over finetuning under tight compute constraints.

### 3.1  DEFINITION OF IN-CONTEXT LEARNING

We consider a general VQA scenario where ICL is performed using a pretrained VLM, denoted as $S$. Let each triplet $(I, Q, A)$ be a data sample consisting of an image $I$ and a question $Q$, and $A$ is the corresponding answer. At inference time, the model is given a test query $\hat{x} = (\hat{I}, \hat{Q}) \sim \mathcal{D}$ to return an answer $\hat{A}$, where $\mathcal{D}$ is the test distribution. ICL involves a context set of length $k$ as $\mathcal{G} = \{(I_i, Q_i, A_i)\}_{i=1}^{k}$, where each entry is a demonstration sampled from a support pool $\mathcal{P}$. We propose that a VLM is capable of performing in-context learning if it achieves lower expected task risk with in-context demonstrations compared to without context demonstrations, i.e.,

$$\mathbb{E}_{\hat{x}\sim\mathcal{D}}\left[\ell(S(\hat{x} \mid \mathcal{G}), \hat{y})\right] < \mathbb{E}_{\hat{x}\sim\mathcal{D}}\left[\ell(S(\hat{x}), \hat{y})\right] + \epsilon, \tag{1}$$

where $\ell$ is a task-specific loss function and $\hat{y}$ is the ground-truth label, and $\epsilon$ is a small negative value controlled by significance test. The left-hand side corresponds to in-context prediction and the right-hand side represents zero-shot prediction. This comparative formulation reflects the effectiveness of ICL is equivalent to improving task performance over zero-shot. While ICL has been shown to be effective in improving performance for LLMs and large VLMs, its effectiveness in small VLMs or even tiny VLMs remains unclear (Zong et al., 2024).

### 3.2  A FEASIBILITY TEST OF SMALL VLMS

We investigate ICL capability of VLMs at different scales in a set of tasks. We first categorize VLMs into tiny (N≤4B), small (4B<N≤12B), medium (12B<N≤40B), and large (N>40B), where N denotes the number of model's parameters. We use a standard image similarity (Zhang et al., 2023b) to select demonstrations. Details of the models tested can be found in Appendix A.1 We consider a challenging ICL reasoning benchmark (VL-ICL (Zong et al., 2024)) and a simpler classification task (CUB (Wah et al., 2011)) for small & tiny

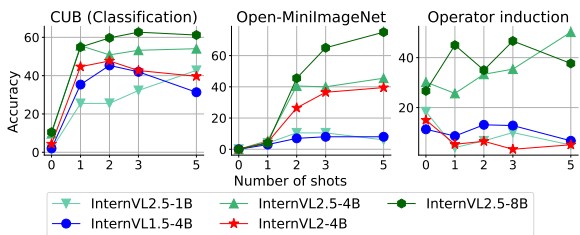

Figure 2: *ICL evaluation for different InternVL sizes and versions. Left to right: tasks are ordered with increasing difficulty. Within the same version (i.e., version 2.5), small VLMs exhibit better ICL capability than tiny VLMs, though tiny VLMs (i.e., size 4B) also improve across versions.*

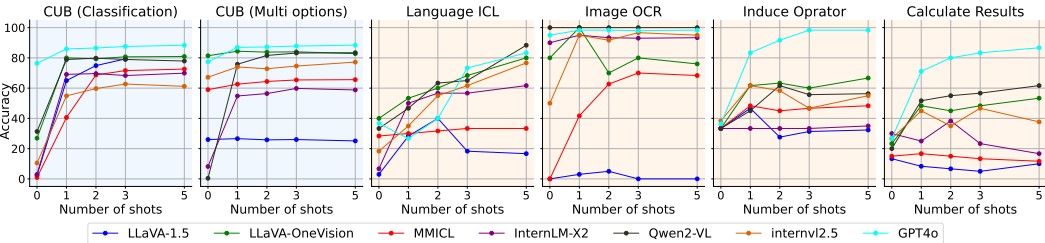

Figure 3: *ICL evaluation to verify models' inherent capabilities (best viewed in color). From left to right: CUB (Classification): image label prediction. CUB (Multi options): (A,B,.. prediction). Language ICL: converted text-only ICL. Image OCR: text recognition from the image. Induce operator: inducing the operator represented by the question mark. Calculate results: final evaluation.*

VLMs to validate our choice of model size. We compare models of the same family to exclude other potential factors and refer to Appendix A.2 for other families. Results are shown in Fig. 2, with tasks ordered from left to right with increasing difficulty. Tiny VLMs struggle to benefit from the demonstrations and to satisfy Eq. 1, especially when the task is more challenging. Meanwhile, we also observe steady improvements in the ICL capability of tiny VLMs across versions (e.g., 4B models), suggesting that ICL may become feasible at this scale with stronger models. In contrast, consistent ICL capability emerges when the model size exceeds 4B. We observe the same property in other model families in Appendix A.2. Therefore, in the rest of the paper we focus on small VLMs, as they exhibit a good balance between efficiency and potential ICL capability.

> **Take-away message #1: What scale?**
>
> In-context reasoning in general scales with size. Currently, small VLMs ($4B < N \le 12B$) achieve a strong balance between efficiency and ICL capability.

**Vision Language ICL (VL ICL) Failure Modes: A Case Study.** We hypothesize that task learning via context can be hierarchically decomposed, with each subtask relying on specific foundational capabilities. For example, in-context classification depends on vision-text alignment, which small VLMs often acquire during pretraining—explaining their success on CUB classification (Fig. 3). However, when reframed as a multi-option VQA, performance drops sharply for LLaVA-1.5 (Fig. 3), likely due to its limited text reasoning ability needed to map image, options, and labels. Examples are provided in Appendix A.3. To identify the sub-capabilities required for VL ICL and to trace its failure modes, we analyze a representative reasoning task from the VL-ICL benchmark: operator induction. We decompose this task into discrete subtasks, each of which must be successfully combined to solve the full problem. We further compare performance on this task when only textual input is provided, highlighting the differences between Vision-ICL and its pure language counterpart.

The operator induction task involves inferring the mathematical operator represented by a question mark in the demonstrations and applying it to a query. For example, given the demonstration *5 ? 8 = 40*, we infer that *?* represents multiplication, and thus the correct answer to the query *7 ? 3* would be 21. Here, we include both in-context tuned VLMs and small powerful models[1]. To better understand the capabilities and limitations of Vision-ICL, we decompose the task into three subtasks that represent the necessary steps for solving the full problem. First, *Image OCR* involves reading the expression from the image, which tests the model's visual perception ability. Second, *Induce Operator* focuses on identifying the operator without performing the final calculation. Third, *Calculate Results* requires computing the final answer based on the predicted operator. Additionally, we convert all inputs to text and require the VLM to reason using only textual input *(Language ICL)*, in order to assess the reasoning capabilities of the underlying LLM after vision-language pretraining. Illustrative examples of these subtasks are provided in Appendix A.6.

With the models we tested, GPT-4o has the largest size and the best performance in almost every step, which serves as an upper-bound reference for small VLMs. In contrast, the behavior of small VLMs significantly varies. The failure at the final step can be traced back to earlier steps: LLaVA-1.5 fails at image OCR, making it impossible to perform the task. Although InternLM-XComposer2 (InternLM-X2) succeeds in the first step, it fails at inducing the operator. Interestingly, the most recent models, such as LLaVA-OneVision, InternVL2.5 and Qwen2-VL exhibit clearly better ICL capability and succeed to improve overall task performance from demonstrations. We believe this superiority

---

[1]Ranking from MMBench

stems from their better overall capability. This is especially clear if we compare LLaVA-1.5 with LLaVA-OneVision, two generations of the same model family. Surprisingly, in-context tuned models (InternLM-X2 and MMICL) fail at inducing the operators from demonstrations. A final interesting observation is that while LLaVA-OneVision, InternVL2.5 and Qwen2-VL rival GPT-4o in pure language ICL, a significant performance gap is exhibited when performing the same task in vision language form. Nevertheless, our findings reveal that recent and modern small VLMs demonstrate promising ICL capabilities, which can be harnessed to boost their performance online.

> **Take-away message #2: VL-ICL challenge**
>
> Multimodal in-context reasoning is more complex, relying on the (successful) combination of core capabilities such as perception, modality alignment, and language reasoning.

**ICL potential for low-resource adaptation.** The inference-only nature of ICL makes it substantially more computationally efficient for injecting new knowledge compared to direct parameter update methods such as fine-tuning (FT). To illustrate this, we compare their performance and compute budgets in Fig. 4. Here, the compute budget is normalized by the total cost of extensive offline fine-tuning under the assumption of ample resources. In practice, additional compute for fine-tuning corresponds to more training epochs, whereas additional compute for ICL corresponds to longer contexts with more demonstrations. Our results show that fine-tuning remains effective for maximizing performance when compute is unconstrained, since ICL does not always scale with longer contexts (Li et al., 2024c). However, under strict compute budgets, ICL delivers markedly greater performance gains than fine-tuning. Furthermore, because ICL requires no explicit

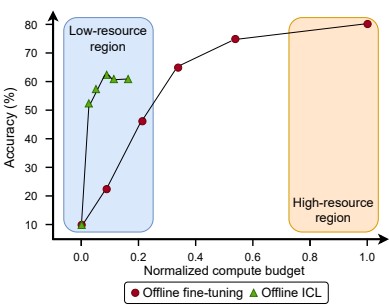

Figure 4: *Comparison of offline fine-tuning v.s. ICL under a compute budget. ICL is a more appealing choice in a low-source environment as it brings significant performance improvement at low computational cost.*

model training, it avoids catastrophic forgetting (French, 1999)—the degradation of general capabilities of the model when learning new tasks. Taken together, these results highlight ICL as a more flexible and robust paradigm for low-resource scenarios.

> **Take-away message #3: Is ICL advantageous for online adaptation?**
>
> Compared to fine-tuning, ICL enables rapid performance improvements with minimal overhead, making it well-suited for online adaptation in low-resource settings.

Having shown that recent small VLMs can significantly benefit from in-context examples offering a clear advantage over direct parameter updates in low-resource settings, we now introduce our method for online adaptation using sparse teacher demonstrations.

## 4 ICD: In-Context Distillation

We consider a real-world deployment setting where test samples are drawn online from the target distribution $\mathcal{D}$ of a deployment environment. In low-resource scenarios, we assume a small student VLM $S$ deployed on edge devices, together with access to a powerful teacher model $T$, a large capable VLM hosted in the cloud. Unlike conventional large-small model knowledge distillation that relies on further training of the small model to improve its performance (Hinton et al., 2015; Hsieh et al., 2023), we propose to transfer the knowledge from the teacher to the student via ICL. We require no efforts to annotate target tasks in advance, rather demonstrations must be constructed directly from the online test stream. To this end, we gather a dynamic demonstration pool $\tilde{\mathcal{P}}$ by querying the teacher model for annotations on the test samples. Formally, $\tilde{\mathcal{P}} = \{(\hat{I}_i, \hat{Q}_i, A'_i)\}_{i=1}^{P}$, where $P$ is the pool size and $A'_i = T(\hat{x}_i)$ denotes the teacher's response. A discussion on the annotation type is provided in Appendix A.5. In this way, the teacher's knowledge is first embedded into the demonstration pool. At inference, the student retrieves a subset of relevant demonstrations $\mathcal{G} \subseteq \tilde{\mathcal{P}}$ as context to enhance the input, enabling the teacher's knowledge to be distilled into the student's predictions without explicit model updates. This collaborative framework of teacher and student depicts the core structure of our in-context distillation (ICD). Inspired by Eq. 1, we formulate the

corresponding optimization problem[2] for our ICD as:

$$\min_{\mathcal{G}} \ \mathbb{E}_{\hat{x} \sim \mathcal{D}} \left[ \ell(S(\hat{x} \mid \mathcal{G}), \hat{y}) \right] - \mathbb{E}_{\hat{x} \sim \mathcal{D}} \left[ \ell(S(\hat{x}), \hat{y}) \right]. \tag{2}$$

Our objective is to construct the best demonstration set $\mathcal{G}$ that maximizes the utility of the context when performing ICL. The advantage of this formulation is multi-fold. First, the use of the teacher model to provide demonstrations alleviates the need for human demonstrations, as they can be expensive and time-consuming, as we show in Tab. 3. Moreover, the dynamic knowledge injection via ICL is training-free, which avoids the computational cost of parameter update and the potential risk of forgetting. Lastly, the asynchronous inference of the student and teacher models helps reduce the latency in online scenarios: the student model does not wait for the teacher model's response in real time, as the teacher's role is only to populate the demonstration pool for later use. To the best of our knowledge, this is the first training-free knowledge distillation framework for VLMs.

In the following, we illustrate our modular design to address the following challenges: (1) the quality of the teacher's annotation; (2) the effectiveness of demonstration selection; and (3) the frequency (and thus the cost) of querying the teacher model.

## 4.1 Improving Annotations with Test-Time Scaling

As a teacher model can still be subject to mistakes and hallucinations, to reduce the noise in our ICD pipeline, we leverage test-time scaling (TTS) techniques to further refine and reduce the noise on the teacher annotations (Muennighoff et al., 2025; Snell et al., 2024). This refinement is beneficial for the student model since ICL for small VLMs is sensitive to the noise in the demonstrations, as shown in Li et al. (2024b). We adopt a simple TTS strategy, namely BestofN (Brown et al., 2024), to improve the reliability of the annotation. Specifically, we perform multiple inferences with the same input and only accept consistent outputs as valid demonstrations.

## 4.2 Improving Demonstration Selection with Cross-Modal Matching

Common practice for selecting demonstrations uses encoders to extract image and text features for all samples, and select demonstrations based on their uni-modal similarity (image to image and query to annotation) to the query (Li et al., 2024b; Baldassini et al., 2024). We observe that while image-based filtering is effective, text based filtering is less reliable due to the generic nature of the query in many cases, e.g., *What species is this bird?* from the CUB dataset (Wah et al., 2011).

To address this, we introduce a cross-modal matching strategy that leverages both image and text (SELECTDEMO in Algo. 2). Given a multimodal encoder $E$, let $F_i, F_t \in \mathbb{R}^{p \times L}$ be the image and text features from the pool $\tilde{\mathcal{P}}$, with $p$ samples and $L$-dimensional embeddings. Each demonstration candidate is encoded as $f_i = E(I)$, $f_t = E(Q, A')$, and the query is encoded as $\hat{f}_i = E(\hat{I})$, $\hat{f}_t = E(\hat{Q})$. We perform a three-step selection in the following order: image-text $\mathcal{R}_{it}(F_t, \hat{f}_i, K_{it})$, image-image $\mathcal{R}_{ii}(F_i, \hat{f}_i, K_{ii})$, and text-text $\mathcal{R}_{tt}(F_t, \hat{f}_t, K_{tt})$, where $\mathcal{R}(\cdot, K)$ denotes top-$K$ filtering on the input features based on simi-

---

**Algorithm 1** Online In-Context Distillation

**Input:** Dataset $\mathcal{D}$, Student $S$, Teacher $T$, Pool $\mathcal{P}$, Multimodal Encoder $E$, Threshold $\delta$
**Output:** Prediction $\hat{A}$ for each samples $x$ in $\mathcal{D}$

1: **for** each $x = (I, Q) \in \mathcal{D}$ **do**
2:    $\hat{A}, u \leftarrow S(x)$
3:    ▷ *Uncertainty check, see Sec.4.3*
4:    **if** $u < \delta$ **then**
5:       **return** $\hat{A}$
6:    **else**
7:       ▷ *Demonstration selection, see Sec.4.2*
8:       $\mathcal{G} \leftarrow \text{SELECTDEMO}(x, \mathcal{P}, E)$
9:       $\hat{A}', u' \leftarrow S(x \mid \mathcal{G})$
10:      **if** $u' < u$ **then**
11:         $\hat{A} \leftarrow \hat{A}'$
12:      **end if**
13:      **if** $u' \geq \delta$ **then**
14:         ▷ *Refine with TTS, see Sec.4.1*
15:         $A' \leftarrow \text{TTS}(T(x))$
16:         $\mathcal{P} \leftarrow \mathcal{P} \cup \{(x, A')\}$
17:      **end if**
18:      **return** $\hat{A}$
19:    **end if**
20: **end for**

---

larity. While a generic query $\hat{Q}$ may limit the utility of $F_t$ in the text-text step $\mathcal{R}_{tt}$, the cross-modal step $\mathcal{R}_{it}$ serves as an effective pre-selection mechanism that compensates for this and leverages the textual information more robustly. Further discussion of the selection mechanism is in Appendix A.15.

---

[2]Note that $\hat{y}$ is not provided but to illustrate the objective.

### 4.3 UNCERTAINTY-AWARE ICL FOR ONLINE EVALUATION

Querying teacher models can still be expensive and time-consuming, especially for the powerful closed-source model. Thus, we extend ICD with an uncertainty-aware mechanism to reduce the annotation rate. For each incoming sample, we estimate the predictive uncertainty $u$ of the student model $S$ with the average entropy of the generation process. We discuss and ablate our design choice in Appendix A.13. Specifically, let $\{p_1, p_2, \ldots, p_J\}$ denote the token-wise probability distributions over the vocabulary $\mathcal{V}$ for a generated output sequence of length $J$, where each $p_j = \text{softmax}(a_j)$ is computed from the model's output logits $a_j$ at decoding step $j$. The model's uncertainty can then be estimated as the average entropy over the output sequence:

$$u = \frac{1}{J} \sum_{j=1}^{J} \mathcal{H}(p_j) = -\frac{1}{J} \sum_{j=1}^{J} \sum_{v \in \mathcal{V}} p_j(v) \log p_j(v). \tag{3}$$

A prediction $\hat{A}$ is accepted as-is if $u$ is less than $\delta$, a threshold calibrated on a small validation set annotated by $T$. Otherwise, we invoke ICL using demonstrations retrieved from $\tilde{\mathcal{P}}$ to obtain a new student prediction $\hat{A}$ and a new uncertainty estimation $u'$. If the uncertainty is reduced $u' < u$, the student outputs $\hat{A}$ ( given the higher likelihood of a correct response). In the case of $u' \geq \delta$ the query is deemed difficult and sent to the cloud for annotation.

Our ICD is illustrated in Fig. 1 (left), and detailed in Algo. 2 and Appendix A.9. We prioritize components that are both simple and effective. This not only improves the computational efficiency but also enhances the accessibility and deployability of our framework for low-resource scenarios. In summary, we present a modular framework with the goal of instant improvement of the student performance at minimal compute/annotation budget combining modules for uncertainty estimation, teacher test-time scaling and cross-modal demonstration selection.

## 5 EXPERIMENTS

**Datasets.** We consider classification-based tasks such as GTSRB (Stallkamp et al., 2012) for traffic sign classification, WikiArt (Tan et al., 2019) for painting style classification, and CUB (Wah et al., 2011) for fine-grained bird classification. Furthermore, we consider the more challenging generative tasks. For instance, we test on Flickr30k (Young et al., 2014) for image captioning, PMC-VQA (Zhang et al., 2023a) for biomedical VQA and VQA-AD (Atakishiyev et al., 2023) for synthetic autonomous driving scenes. In Appendix A.11, we discuss our choice of datasets as opposed to more generic ones such as TextVQA (Singh et al., 2019) and OK-VQA (Marino et al., 2019). Details of prompts used for each dataset are in Appendix A.7.

**Models.** We choose two representative models that have been shown effective in in-context reasoning from our feasibility tests in Sec. 3, i.e., LLaVA-OneVision and InternVL2.5. Complete tables with more models can be found in Appendix A.10. Unless otherwise stated, we consider the small VLMs (4B<N≤12B) as students, and GPT-4o (Achiam et al., 2023) as the teacher.

**Baselines.** For self-boosting methods applied to the student model, we include chain-of-thought (*CoT*) prompting (Wei et al., 2022b) and *BestofN* (Brown et al., 2024) as representative test-time scaling techniques. To assess the utility of the teacher annotations, we consider *self-labeling*, similar to the self-generation in (Chen et al., 2023a) for LLMs. In *self-labeling* the pool of demonstration pool is updated with the student model's predictions. We refer to our method as ICD and we consider the *offline ICD* variant where the pool is pre-collected and fully annotated in advance. A detailed discussion of the impact of different sizes of the offline pool is provided in Appendix A.12. We also compare with other VL ICL methods in Sec. 5.2.

**Evaluation metrics.** We evaluate 3-shot ICL by default (full tables with different shots in Appendix A.10). Accuracy on the test set is used for classification and VQA tasks. For image captioning, we reported the BLEU-4 (Papineni et al., 2002) scores. We report the teacher's annotation rate $T(x)\%$ as the percentage of requested annotations given the size of the test stream. More details on implementation and inference computational costs are in Appendix A.4.

### 5.1 MAIN RESULTS

In Table 1, we compare different methods on various tasks. Self-boosting methods (i.e., CoT and BestofN) struggle to achieve consistent and significant improvements over 0-shot performance.

Table 1: *Online evaluation results on various datasets. Our ICD method consistently outperforms zero-shot baselines and prior approaches. "LLaVA" refers to LLaVA-OneVision, and "InternVL" refers to InternVL2.5. T(x) denotes the frequency of querying the teacher model. The best results of each model are highlighted in bold. Results of GPT-4o and offline ICD are included as reference.*

| | GTSRB | | CUB | | WikiArt | | Flickr30k | | PMC-VQA | | VQA-AD | | Average | |
|---|---|---|---|---|---|---|---|---|---|---|---|---|---|---|
| | LLaVA | InternVL | LLaVA | InternVL | LLaVA | InternVL | LLaVA | InternVL | LLaVA | InternVL | LLaVA | InternVL | LLaVA | InternVL |
| 0-shot | 42.6 | 47.2 | 26.9 | 10.5 | 43.1 | 34.1 | 26.7 | 10.7 | 47.5 | 34.5 | 37.0 | 19.0 | 37.3 | 26.0 |
| CoT | 32.1 | 41.0 | 18.9 | 12.8 | 36.5 | 35.9 | 20.5 | 13.5 | 48.2 | 30.1 | 28.0 | 24.0 | 30.7 | 26.2 |
| | -10.5 | -7.2 | -8.0 | +2.3 | -6.6 | +1.8 | -6.2 | +2.8 | +0.7 | -4.4 | -9.0 | +5.0 | -6.6 | +0.2 |
| BestofN | 44.2 | 50.4 | 29.8 | 14.2 | 43.6 | 34.4 | 26.0 | 10.2 | 48.7 | 34.7 | 42.0 | 23.0 | 39.1 | 27.8 |
| | +1.6 | +3.2 | +2.9 | +3.7 | +0.5 | +0.3 | -0.7 | -0.5 | +1.2 | +0.2 | +5.0 | +4.0 | +1.8 | +1.8 |
| Self-labeling | 41.6 | 47.7 | 27.2 | 11.8 | 40.4 | 32.5 | 24.7 | 11.2 | 48.0 | 35.1 | 41.0 | 21.0 | 37.2 | 26.6 |
| | -1.0 | +0.5 | +0.3 | +1.3 | -2.7 | -1.6 | -2.0 | +0.5 | +0.5 | +0.6 | +4.0 | +2.0 | -0.1 | +0.6 |
| ICD | **70.8** | **61.9** | **60.3** | **56.5** | **49.0** | **41.3** | **27.3** | **13.7** | **51.1** | **38.1** | **47.0** | **33.0** | **50.9** | **40.8** |
| | +28.2 | +14.7 | +33.4 | +46.0 | +5.9 | +7.2 | +0.6 | +3.0 | +3.6 | +3.6 | +10.0 | +14.0 | +13.6 | +14.8 |
| ↪ *T(x)* | *4.4* | *6.8* | *4.2* | *6.8* | *20.4* | *11.1* | *21.3* | *18.3* | *39.3* | *31.1* | *18.2* | *14.5* | *17.9* | *14.7* |
| Offline ICD | 72.6 | 66.1 | 73.1 | 62.9 | 53.3 | 44.1 | 26.9 | 14.1 | 48.5 | 38.9 | 50.0 | 37.0 | 54.0 | 43.8 |
| ↪ *T(x)* | *100.0* | *100.0* | *100.0* | *100.0* | *100.0* | *100.0* | *100.0* | *100.0* | *100.0* | *100.0* | *100.0* | *100.0* | *100.0* | *100.0* |
| GPT-4o | 68.1 | | 76.4 | | 56.3 | | 24.6 | | 55.8 | | 41.0 | | 53.7 | |

While such refinements reduce uncertainty and randomness in student predictions, they do not fundamentally enhance the model's capability. Similar limitations arise with self-labeling: despite access to additional data at inference time, it contributes little new information in unseen domains. These observations highlight the necessity of leveraging a strong teacher model as a source of external knowledge to improve student performance. Importantly, our ICD uses the teacher model in a restrained manner through uncertainty thresholding, while achieving close performance to a fully annotated offline ICD. For instance, ICD boosts student performance from 42.6% to 70.8%, compared to zero-shot or TTS-based baselines with as little as 4.4% annotation rate, more importantly outperforming GPT-4o performance (68.1%).

Different tasks present varying levels of difficulty in this scenario. Classification tasks show the good potential of ICL when necessary capabilities are fulfilled, i.e. image-text alignments, as we analyze in Sec. 3.2. Comparing our proposed framework ICD to the model's zero-shot, we observe a consistent boost in performance, which closes the gap between the small VLMs and the powerful teacher, e.g. we enhance LLaVA-OneVision and InternVL2.5 from lower than 30% to around 60% on the CUB dataset, which is significantly closer to 76.4% of GPT-4o. This performance boost enables the model to be practically beneficial in these tasks with no change in the model's parameters. Interestingly, while the teacher model (GPT-4o) does not achieve a better BLEU-4 score than the student LLaVA-OneVision in the image captioning task (24.6 v.s. 26.7 on Flickr30k), it still helps the model to improve via ICL compared to self-labeling. We speculate this is because GPT-4o tends to generate more detailed captions. To validate this, we calculate the string length of each model's caption. While LLaVA-OneVision's average length is 44.2 characters, that of GPT-4o is 62.6, which is 40% longer. These two cases demonstrate that the teacher with a more consistent labeling strategy, especially with TTS, and more detailed answers would provide additional information and reduce the uncertainty of the student. **On average, our ICD brings 14.8% and 13.6% points of performance gain for InternVL2.5 and LLaVA-OneVision across all tasks**, respectively.

## 5.2 ADDITIONAL ANALYSIS

**Computational analysis.** We emphasize the efficiency of our method in achieving strong performance with minimal computational cost, compared with learning-based methods. To estimate cost, we use widely adopted Python packages—`CodeCarbon` and `EcoLogits`—to track carbon emissions during training and inference. These measures are simple to compute and serve as reliable proxies for compute overhead. We compare our ICD approach with LoRA (Hu et al., 2022) fine-tuning in an online learning setting designed for low-resource scenarios. Specifically, we report cumulative carbon emissions and evaluation accuracy after every 64 new annotations obtained

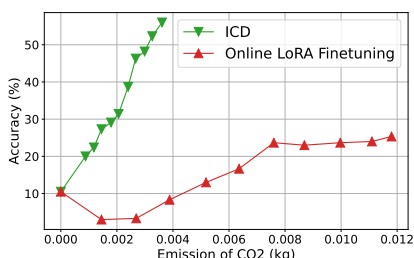

Figure 5: *Accumulative accuracy vs. compute cost in an online scenario. Our ICD excels at this tradeoff.*

from the teacher model, which together form a mini-batch for online learning. Details of online learning can be found in Appendix A.16. Fig. 5 shows that ICD provides an instant performance

boost without the burden of unstable online model training. Moreover, ICD avoids the usual trade-off between compute and performance, delivering the highest performance at minimal cost.

**ICL selection mechanism.** Using LLaVA-OneVision as the base model, we compare with several existing ICL selection strategies: VICL (Zhou et al., 2024b), PICa (Yang et al., 2022), RICES (Alayrac et al., 2022; Awadalla et al., 2023), SQ, SQ-I (Li et al., 2024b) and MMICES (Chen et al., 2025a). Implementation details of these methods can be found in Appendix A.4. Results are presented in Tab. 2 under the offline ICD setting to guarantee enough demonstrations for selection analysis. Our cross-modal selection consistently achieves the best performance across datasets.

Table 2: *Comparison with different ICL methods with the teacher model's annotation in offline setting. Our cross-modal selection achieves the best performance across datasets.*

| Method | | CUB | GTSRB | VQA-AD |
|---|---|---|---|---|
| 0-shot | | 26.9 | 42.6 | 37.0 |
| **ICL Method** | **ICL details** | | | |
| ICD | Our cross-modal selection | **73.1** | **72.6** | **50.0** |
| VICL | Image to caption | 65.2 | 65.2 | 43.0 |
| PICa | Image to caption & tags | 63.5 | 61.4 | 40.0 |
| RICES | Image-image selection | 72.6 | 69.3 | 47.0 |
| SQ | Text-text selection | 26.7 | 39.8 | 47.0 |
| SQ-I | $\mathcal{R}_{tt} + \mathcal{R}_{ii}$ | 68.7 | 70.0 | 44.0 |
| MMICES | $\mathcal{R}_{ii} + \mathcal{R}_{tt}$ | 72.9 | 70.8 | 48.0 |

**Teacher model generality.** Fig. 6 reports the performance gain with different teacher models to highlight the flexibility of our framework. Overall, ICD succeeds in improving the student performance and brings it closer to the corresponding teacher performance, a consistent significant improvement over the student 0-shot performance. Moreover, open-source medium VLMs, which can be locally deployed, serve as an efficient alternative to closed-source mod-

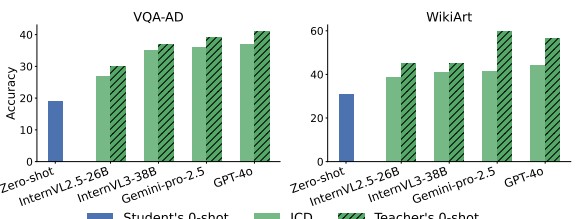

Figure 6: *ICD with different teacher models. We report teacher's 0-shot performance for reference. ICD brings the performance of the student model closer to that of the teacher.*

els. For instance, InternVL3-38B achieves a comparable performance boost to Gemini-pro-2.5 in both VQA-AD and WikiArt. Notably, our training-free ICD provides significant improvement across different teacher models, showing a flexibility in framework design.

**Practical costs.** We include a rough estimation of cost for a WikiArt-scale dataset of 80k images in Tab 3, details are in Appendix A.8. While human annotation is thorough and invaluable, it can be time-consuming. In contrast, automatic annotation can generally be completed within one day. The cost of model annotation is also comparably low. As a reference, annotat-

Table 3: *Practical costs with different annotation strategies. ICD balances well between cost and performance.*

| Baseline | Source of Annotation | Cost | Time | WikiArt | |
|---|---|---|---|---|---|
| | | | | 0-shot | 3-shot |
| | Human | >4000 $ | 40 days | | 44.7 |
| | Self | < 1 $ | 4 hours | | 32.5 |
| InternVL-2.5 | GPT-4o | < 40 $ | 20 hours | 34.1 | 44.1 |
| | InternVL-26B | < 1 $ | 8 hours | | 38.6 |

ing an artistic style dataset (Ruta et al., 2022) of 135k images costs in total 160k $. Overall, our ICD strikes a good balance between performance and practical costs.

## 6 CONCLUSION

This work explores the potential of enabling small VLMs to perform effectively under low compute budgets. We propose a simple yet effective in-context distillation (ICD) framework to enhance small VLMs, eliminating the need for human annotation and model retraining. Our approach enhances the large teacher model's annotation with test-time scaling, and provides the small student model with a dynamically populated demonstration pool for in-context learning, incorporating cross-modal demonstration selection and uncertainty-aware querying. We also performed an in-depth analysis of ICL, revealing key factors for ICL effectiveness in small VLMs. We validate our ICD approach via a wide range of experiments, in a realistic and challenging online evaluation, showing strong performance gains at low annotation and compute costs. To the best of our knowledge, we are the first to explore an effective annotation- and training-free framework for large to small VLMs distillation. Limitations of our proposed method are discussed in Appendix A.14.

**Ethics Statement.** The authors confirm that they have read and commit to adhering to the ICLR Code of Ethics.

**Reproducibility statement.** We provide all necessary implementation details to ensure reproducibility, including the choice of models, the text prompts used for each dataset and task, and detailed descriptions of the techniques employed in this work in the main paper and Appendix. The source code would be released upon acceptance.

**Large Language Model Assistance.** Large language models were used to check and correct grammatical aspects. The authors have thoroughly reviewed and edited all content and take full responsibility for the final published work.

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

## A  APPENDIX

### A.1  MODELS CARDS

In this section, we list the models that we tested in this work in Tab. 4. Unless specified, we refer to the small VLMs (4B<N≤12B) when calling a model family. For instance, when we use the term LLaVA-OneVision, it refers to LLaVA-OneVision-7B.

Table 4: List of different models tested in our work.

| Model Name | Model Size | Category | Link |
|---|---|---|---|
| LLaVA-OneVision | 0.5B | Tiny | link |
| InternVL2.5 | 1B | Tiny | link |
| InternVL2.5 | 2B | Tiny | link |
| InternVL2.5 | 4B | Tiny | link |
| Qwen2-VL | 2B | Tiny | link |
| InternVL2.5 | 8B | Small | link |
| LLaVA-1.5 | 7B | Small | link |
| LLaVA-OneVision | 7B | Small | link |
| Qwen2-VL | 7B | Small | link |
| InternLM-XComposer2 | 7B | Small | link |
| MMICL | 12B | Small | link |
| InternVL2.5 | 26B | Medium | link |
| GPT-4o | Unknown | Large | link |

### A.2  ANALYSIS OF THE MODEL SIZE

In Fig. 7, we present the complete comparison with three different model families: InternVL2.5, LLaVA-OneVision, and Qwen2-VL. We observe the same trend: small models benefit from ICL while tiny VLMs struggle to improve with demonstrations.

### A.3  VISUALIZATION OF CUB EXPERIMENTS

In this section, we present illustrative examples for the two variants of the VQA task based on the CUB dataset: `CUB (Classification)` and `CUB (Multi Options)`.

- **CUB (Classification)**

  The prompts follow the standard conversion of classification tasks to VQA tasks, as shown in Sec. A.7. An illustrative example is in Fig. 8

- **CUB (Multi Options)**

  For multi-option classification, we just take the ground truth class label and randomly sample 3 other options to form a 4-option question. An illustrative example is in Fig. 9.

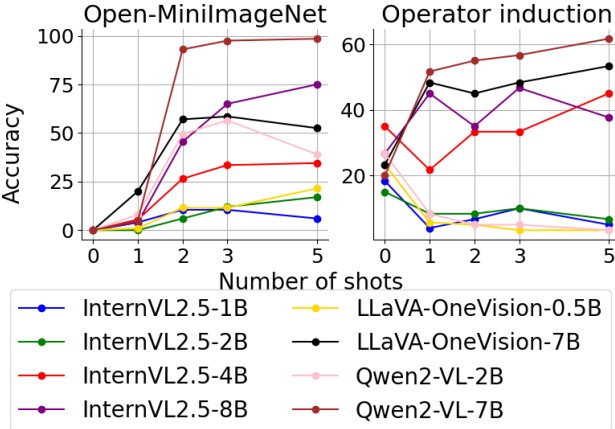

Figure 7: ICL evaluation with respect to the model size. From left to right, the tasks are more challenging to perform ICL. Small VLMs benefit better from emerging ICL capability than tiny VLMs.

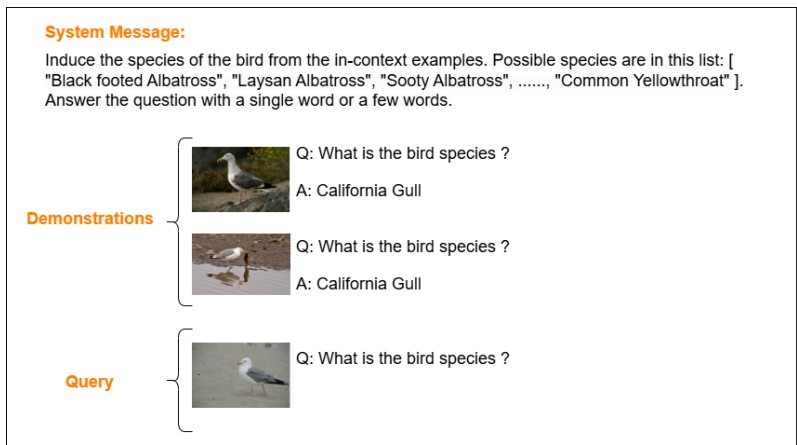

Figure 8: Example of ICL CUB (Classification) task. The query sample is prepended with 2 demonstrations in this example.

### A.4 IMPLEMENTATION AND COMPUTATION DETAILS

In this section, we present the details of implementation and the discussion of the computation cost of our proposed method.

### A.4.1 IMPLEMENTATION

**Data.** We use the training set of each dataset as the source of the unlabeled dataset, and also to compare the quality of teacher annotations with human annotations.

**Models.** We use the open-source models (as listed in Appendix A.1) that are available and can be downloaded from the Hugging Face repository. Each model is loaded and set to the `eval` mode for inference. Unless specified, there are no model training or parameter updates on any models.

**Inference** The student models are tested with zero-shot or in-context learning with the query samples (and demonstrations), which are formatted into an input prompt as described in Appendix A.7. As for selecting demonstrations, we used $K_{it} = 50\% \times |\hat{\mathcal{P}}|$ where $|\hat{\mathcal{P}}|$ is the number of samples in the pool. Moreover, $K_{ii} = 10$ to select the top 10 similar samples with image similarities and $K_{tt} = 5$ to select the samples as demonstration.

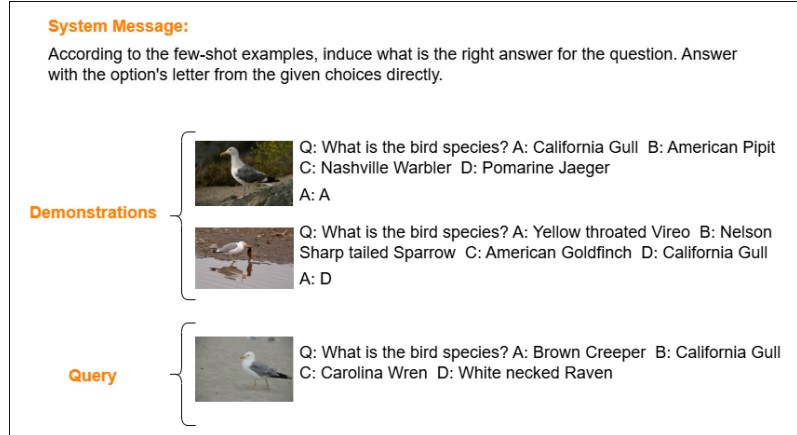

Figure 9: Example of ICL CUB (Multi Options) task. The query sample is prepended with 2 demonstrations in this example.

**Annotation** The teacher model annotates the provided sample with the same prompts presented in Appendix A.7. To further refine the annotation, we apply test-time scaling techniques to reduce the noise and uncertainty in the prediction. We forward 3 times the same input to GPT-4o with its default *temperature* = 1 and keep only the consistent predictions. Specifically, for short text such as multi-option VQA or classification, we can simply compare the consistency of the outputs using an exact match. For captioning tasks, we first calculate the BLEU-2 score between each pair of the three predictions. If they are all high in general ($> 0.5$ in practice), we regard this annotation as consistent.

**Online Pipeline** The threshold of the uncertainty measure plays an important role in our selective annotation and ICL. In practice, we found that the threshold can be transferred among different tasks. Specifically, for CUB, WikiArt, GTSRB, PMC-VQA, and VQA-AD, the threshold is set to be 0.4. Instead, for Flickr30k, the threshold is set to be 1.6.

**Baseline ICL methods.** To ensure a fair comparison, all ICL methods use the same multi-modal encoder (i.e., SigLIP: `google/siglip-so400m-patch14-384`) rather than the standard CLIP model in PICa, SQ and SQ-I. This more powerful encoder leads to better demonstration selection. For instance, SQ-I [25] achieves 68.7% with SigLIP vs. 56.2% with CLIP on CUB.

**Code** We developed our method, analysis, and online pipeline based on the implementation of VL-ICLZong et al. (2024) benchmark.

### A.4.2 COMPUTATION COST

In this section, we discuss the computation cost of our proposed framework in two aspects. First, we discuss the computation at inference time that is related to ICL. Moreover, we discuss the broader ecological impact of our proposed method with the computation we can save for an environmentally friendly AI.

**Inference-Time Computation** We report average inference time under consistent configurations as a proxy for computational overhead. Zero-shot evaluation involves a single forward pass, whereas ICL introduces two additional costs. First, for each query, features must be

Table 5: Average inference time for each sample.

| Mode | 0-shot | 1-shot | 2-shot | 3-shot | 5-shot |
|---|---|---|---|---|---|
| **Time (s)** | 0.375 | 1.325 | 1.610 | 1.895 | 2.490 |

extracted and used to retrieve relevant demonstrations—this involves a forward pass through the multi-modal encoder and one or more matrix multiplications to calculate similarity and identify top candidates (see Sec. 4). This retrieval step is performed only once per query. Second, the inclusion of demonstrations increases the total number of input tokens, adding overhead during inference. As shown in Tab. 5, the jump from 0-shot to 1-shot yields the largest increase in computation time, while

the cost scales approximately linearly from 1-shot to 5-shot. All experiments are conducted on a single NVIDIA RTX A6000 GPU with 48 GB of memory.

**Computational Efficiency and Ecological Impact**   Our method is entirely training-free, eliminating the need for computationally intensive fine-tuning, which significantly reduces energy consumption and environmental impact. In contrast to full model retraining, our approach leverages inference-time adaptation through ICL, avoiding repeated gradient-based optimization. Furthermore, we do not query the large teacher model for every test sample. Instead, annotation is performed selectively, conditioned on the student model's uncertainty. This conditional querying strategy substantially reduces the number of teacher invocations, as we show in Tab. 1 of the main paper, resulting in additional computational savings and improved sustainability.

## A.5   TEACHER'S ANNOTATION

While annotating the data with labels is a common practice, we can also ask the teacher to generate other annotations in other forms, as shown in Tab. 6. Additional descriptions bring minor improvement, as they might be complementary to small details that a small VLM cannot perceive. Furthermore, distilling the chain of thought (CoT) in addition to the label provides an additional boost. We conjecture that it triggers the reasoning ability of the small models and reduces the risk of shortcut learning. We leave the exploration of more sophisticated teacher-annotation paradigms for future work.

Table 6: Comparison of different annotation types from the teacher model.

| Baseline | Annotation | CUB | | | | |
|---|---|---|---|---|---|---|
| | | Zero shot | 1 shot | 2 shot | 3 shot | 5 shot |
| LLaVA-OneVision | Label | 26.9 | 72.4 | 73.1 | 73.1 | 72.9 |
| | + Description | | 72.6 | 73.2 | 73.3 | 72.8 |
| | + CoT | | 72.7 | 74.0 | 73.7 | 73.0 |

## A.6   VISUALIZATION OF OPERATOR INDUCTION TASK

In this section, we visualize the original Operator Induction task as in Fig. 10. This is also the Calculate Results task in Sec. 3.2, which requires the model to output the final results. Other sub-tasks are presented as follows.

- Text Reasoning: the model is required to infer the operator and calculate the results with purely text input, as shown in Fig. 11
- Image OCR: the model is required to do text recognition to read the expression from the image, as shown in Fig. 12
- Induce Operator: the model is required to infer the operator without calculating the results. Legal answers are provided to help the model better reason, as shown in Fig. 13

## A.7   PROMPTS FOR EACH DATASET

In this section, we list the prompts we used for each dataset. We mostly follow the implementation of VL-ICLZong et al. (2024) while slightly modifying some texts to improve the overall performance. The general form of the prompt is as follows:

```
<System message>

# Concatenation of demonstrations
<Demonstration Image>
<Demonstration Question>
<Demonstration Answer>
```

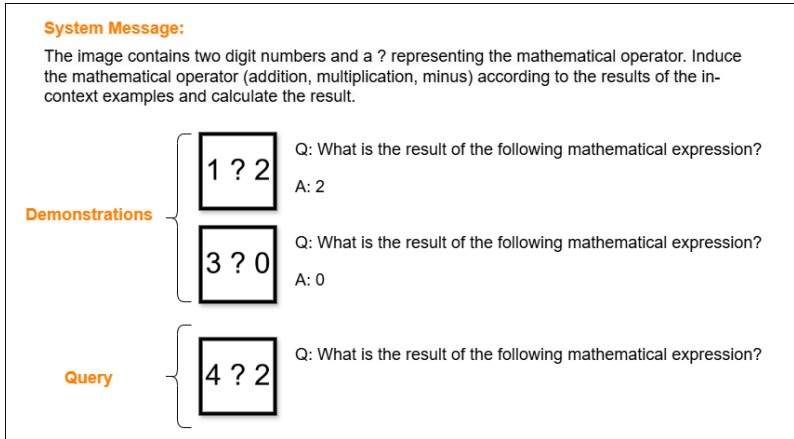

Figure 10: Example of ICL for operator induction task. The query sample is prepended with 2 demonstrations in this example.

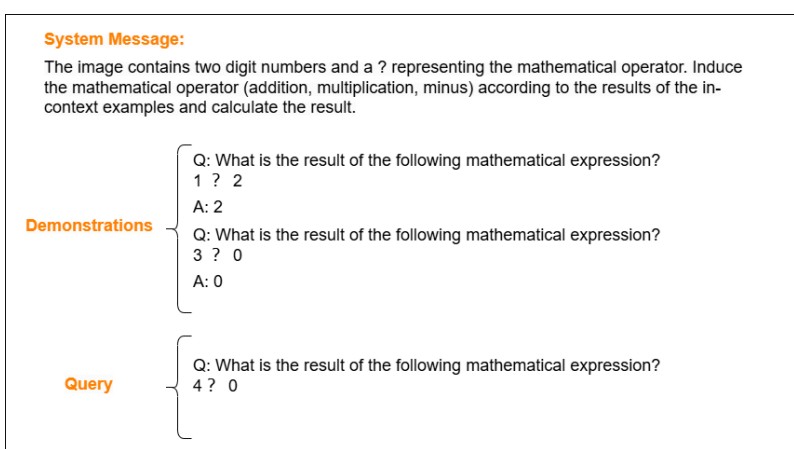

Figure 11: Example of ICL for operator induction **subtask**: Text Reasoning. The query sample is prepended with 2 demonstrations in this example.

```
<Query Image>
<Query Question>
```

The structure might vary slightly for the requirements of each model. We present the standardized question and system message we provide to the models for each task.

### A.7.1 QUESTION FOR EACH TASK

- **GTSRB Stallkamp et al. (2012)**
  *Question: What is the traffic sign?*
- **WikiArt Tan et al. (2019)**
  *Question: What is the painting style?*
- **CUB Wah et al. (2011)**
  *Question: What is the bird species ?*
- **PMC-VQA Zhang et al. (2023a)**
  *Question (with options): What is observed in the control nuclei from P. australis roots in panel (d)? A: Chromatin condensation B: Fragmentation of nuclei C: Lignin labeling D: None of the above.*

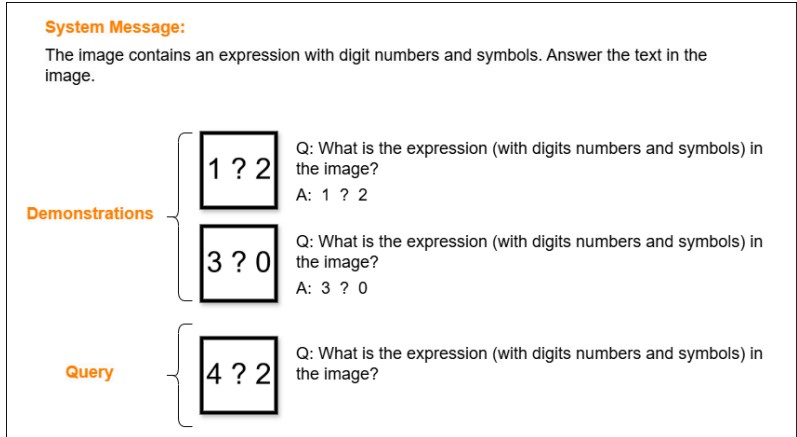

Figure 12: Example of ICL for operator induction **subtask**: Image OCR. The query sample is prepended with 2 demonstrations in this example.

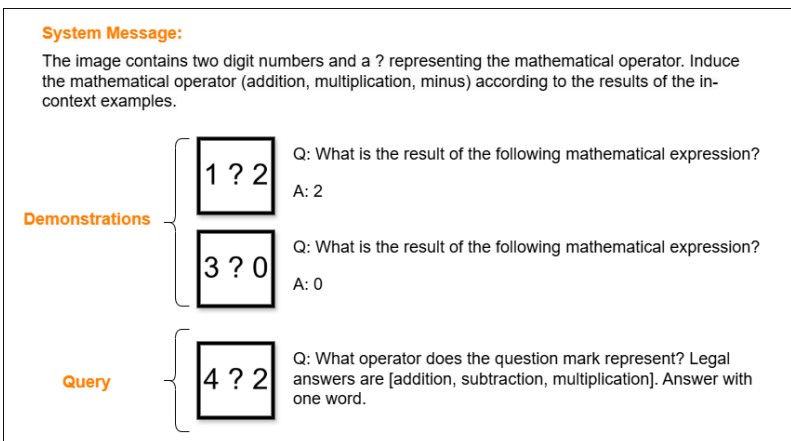

Figure 13: Example of ICL for operator induction **subtask**: Induce Operator. The query sample is prepended with 2 demonstrations in this example.

- **VQA-AD Atakishiyev et al. (2023)**

  *Question: What will be the next action of the car?*

- **Flickr30k Young et al. (2014)**

  *Describe this image in a single sentence.*

A.7.2 SYSTEM MESSAGE

- **GTSRB Stallkamp et al. (2012)**

```
Induce the traffic sign from the in-context examples.
Possible answers are in this list:  ["Speed limit (20
km/h)", "Speed limit (30 km/h)", "Speed limit (50 km/h)",
"Speed limit (60 km/h)", "Speed limit (70 km/h)", "Speed
limit (80 km/h)", "End of speed limit (80 km/h)", "Speed
limit (100 km/h)", "Speed limit (120 km/h)", "No passing",
"No passing for vehicles over 3.5 tonnes", "Right-of-way at
the next intersection", "Priority road", "Yield", "Stop",
"No vehicles", "Vehicles over 3.5 tonnes prohibited",
"No entry", "General caution", "Dangerous curve to the
left", "Dangerous curve to the right", "Double curve",
"Bumpy road", "Slippery road", "Road narrows on the right",
```

"Road work", "Traffic signals", "Pedestrians", "Children
crossing", "Bicycles crossing", "Beware of ice/snow", "Wild
animals crossing", "End of all speed and passing limits",
"Turn right ahead", "Turn left ahead", "Ahead only", "Go
straight or right", "Go straight or left", "Keep right",
"Keep left", "Roundabout mandatory", "End of no passing",
"End of no passing for vehicles over 3.5 tonnes"]. Answer
the question with a single word or a few words.

- **WikiArt Tan et al. (2019)**

  Induce the painting style from the in-context examples.
  Possible styles are in this list: ['Expressionism',
  'Pop Art', 'High Renaissance', 'Symbolism', 'Minimalism',
  'Pointillism', 'Action painting', 'Abstract Expressionism',
  'Naive Art Primitivism', 'New Realism', 'Impressionism',
  'Post Impressionism', 'Mannerism Late Renaissance',
  'Analytical Cubism', 'Northern Renaissance', 'Contemporary
  Realism', 'Romanticism', 'Baroque', 'Ukiyo e', 'Early
  Renaissance', 'Realism', 'Fauvism', 'Art Nouveau Modern',
  'Color Field Painting', 'Rococo', 'Cubism', 'Synthetic
  Cubism']. Answer the question with a single word or a few
  words.

- **CUB Wah et al. (2011)**

  Induce the species of the bird from the in-context examples.
  Possible species are in this list: [ "Black footed
  Albatross", "Laysan Albatross", "Sooty Albatross", "Groove
  billed Ani", "Crested Auklet", "Least Auklet", "Parakeet
  Auklet", "Rhinoceros Auklet", "Brewer Blackbird", "Red
  winged Blackbird", "Rusty Blackbird", "Yellow headed
  Blackbird", "Bobolink", "Indigo Bunting", "Lazuli Bunting",
  "Painted Bunting", "Cardinal", "Spotted Catbird", "Gray
  Catbird", "Yellow breasted Chat", "Eastern Towhee", "Chuck
  will Widow", "Brandt Cormorant", "Red faced Cormorant",
  "Pelagic Cormorant", "Bronzed Cowbird", "Shiny Cowbird",
  "Brown Creeper", "American Crow", "Fish Crow", "Black
  billed Cuckoo", "Mangrove Cuckoo", "Yellow billed Cuckoo",
  "Gray crowned Rosy Finch", "Purple Finch", "Northern
  Flicker", "Acadian Flycatcher", "Great Crested Flycatcher",
  "Least Flycatcher", "Olive sided Flycatcher", "Scissor
  tailed Flycatcher", "Vermilion Flycatcher", "Yellow
  bellied Flycatcher", "Frigatebird", "Northern Fulmar",
  "Gadwall", "American Goldfinch", "European Goldfinch",
  "Boat tailed Grackle", "Eared Grebe", "Horned Grebe", "Pied
  billed Grebe", "Western Grebe", "Blue Grosbeak", "Evening
  Grosbeak", "Pine Grosbeak", "Rose breasted Grosbeak",
  "Pigeon Guillemot", "California Gull", "Glaucous winged
  Gull", "Heermann Gull", "Herring Gull", "Ivory Gull",
  "Ring billed Gull", "Slaty backed Gull", "Western Gull",
  "Anna Hummingbird", "Ruby throated Hummingbird", "Rufous
  Hummingbird", "Green Violetear", "Long tailed Jaeger",
  "Pomarine Jaeger", "Blue Jay", "Florida Jay", "Green Jay",
  "Dark eyed Junco", "Tropical Kingbird", "Gray Kingbird",
  "Belted Kingfisher", "Green Kingfisher", "Pied Kingfisher",
  "Ringed Kingfisher", "White breasted Kingfisher", "Red
  legged Kittiwake", "Horned Lark", "Pacific Loon", "Mallard",
  "Western Meadowlark", "Hooded Merganser", "Red breasted
  Merganser", "Mockingbird", "Nighthawk", "Clark Nutcracker",
  "White breasted Nuthatch", "Baltimore Oriole", "Hooded
  Oriole", "Orchard Oriole", "Scott Oriole", "Ovenbird",

```
"Brown Pelican", "White Pelican", "Western Wood Pewee",
"Sayornis", "American Pipit", "Whip poor Will", "Horned
Puffin", "Common Raven", "White necked Raven", "American
Redstart", "Geococcyx", "Loggerhead Shrike", "Great
Grey Shrike", "Baird Sparrow", "Black throated Sparrow",
"Brewhars Sparrow", "Chipping Sparrow", "Clay colored
Sparrow", "House Sparrow", "Field Sparrow", "Fox Sparrow",
"Grasshopper Sparrow", "Harris Sparrow", "Henslow Sparrow",
"Le Conte Sparrow", "Lincoln Sparrow", "Nelson Sharp
tailed Sparrow", "Savannah Sparrow", "Seaside Sparrow",
"Song Sparrow", "Tree Sparrow", "Vesper Sparrow", "White
crowned Sparrow", "White throated Sparrow", "Cape Glossy
Starling", "Bank Swallow", "Barn Swallow", "Cliff Swallow",
"Tree Swallow", "Scarlet Tanager", "Summer Tanager",
"Artic Tern", "Black Tern", "Caspian Tern", "Common Tern",
"Elegant Tern", "Forsters Tern", "Least Tern", "Green
tailed Towhee", "Brown Thrasher", "Sage Thrasher", "Black
capped Vireo", "Blue headed Vireo", "Philadelphia Vireo",
"Red eyed Vireo", "Warbling Vireo", "White eyed Vireo",
"Yellow throated Vireo", "Bay breasted Warbler", "Black
and white Warbler", "Black throated Blue Warbler", "Blue
winged Warbler", "Canada Warbler", "Cape May Warbler",
"Cerulean Warbler", "Chestnut sided Warbler", "Golden
winged Warbler", "Hooded Warbler", "Kentucky Warbler",
"Magnolia Warbler", "Mourning Warbler", "Myrtle Warbler",
"Nashville Warbler", "Orange crowned Warbler", "Palm
Warbler", "Pine Warbler", "Prairie Warbler", "Prothonotary
Warbler", "Swainson Warbler", "Tennessee Warbler", "Wilson
Warbler", "Worm eating Warbler", "Yellow Warbler", "Northern
Waterthrush", "Louisiana Waterthrush", "Bohemian Waxwing",
"Cedar Waxwing", "American Three toed Woodpecker", "Pileated
Woodpecker", "Red bellied Woodpecker", "Red cockaded
Woodpecker", "Red headed Woodpecker", "Downy Woodpecker",
"Bewicks Wren", "Cactus Wren", "Carolina Wren", "House
Wren", "Marsh Wren", "Rock Wren", "Winter Wren", "Common
Yellowthroat" ]. Answer the question with a single word or
a few words.
```

- **PMC-VQA Zhang et al. (2023a)**

  ```
  According to the few-shot examples, induce what is the right
  answer for the question. Answer with the option's letter
  from the given choices directly.
  ```

- **VQA-AD Atakishiyev et al. (2023)**

  ```
  According to the few-shot examples, induce what the right
  answer is for the question. Answer with the option's letter
  from the given choices directly.
  ```

- **Flickr30k Young et al. (2014)**

  ```
  Induce the concept from the in-context examples. Answer
  the question with a single sentence that best captions the
  image.
  ```

A.8  COST AND TIME ESTIMATION FOR DATA ANNOTATION

We analyze the cost and time required for annotating 80k images using three different strategies: (1) OpenAI's GPT-4o API, (2) locally hosted InternVL2.5 models (InternVL2.5-7B and InternVL2.5-26B), and (3) human annotation.

### A.8.1 COST AND TIME ESTIMATION FOR GPT-4O API

GPT-4o API pricing is based on token usage, where input tokens cost 2.50 $ per million tokens and output tokens cost 10.00 $ per million tokens (by February 2025). Assuming a text for input prompt of 50 tokens for annotation instructions and an input of 100 tokens per image with `High` resolution, and the output prompt for a few words as annotation with 10 tokens.

The corresponding cost is calculated as:

$$C_{input} = \frac{(100 + 50) * 80000}{1,000,000} \times 2.50 = 30 \ \$ \tag{4}$$

$$C_{output} = \frac{10 * 80,000}{1,000,000} \times 10.00 = 8.00 \ \$ \tag{5}$$

$$C_{total} = C_{input} + C_{output} = 38 \ \$ \tag{6}$$

In practice, the process and communication time for one image with GPT-4o-API is around 1 second:

$$T_{total} = 80,000 \ \text{sec} \approx 20 \ \text{hours} \tag{7}$$

### A.8.2 COST AND TIME ESTIMATION FOR INTERNVL2.5 MODELS ON RTX A6000

For a locally hosted InternVL2.5-8B model on an RTX A6000 GPU, the average inference time per sample is empirically estimated as 0.2 seconds.

For 80,000 images, the total processing time is:

$$T_{total} = 80,000 \times 0.2 \approx 4 \ \text{hours} \tag{8}$$

The power consumption of an RTX A6000 system is estimated at 0.4 kW, leading to the following energy costs at 0.15 $ per kWh:

$$C_{total} = 0.4 \times 4 \times 0.15 = 0.24 \ \$ \tag{9}$$

Similarly, for a medium-size VLM InternVL2.5-26B, the processing time for each image is empirically estimated as 0.4 seconds. Therefore, the time and cost would be a factor of 2 of that of InternVL2.5-8B. Thus,

$$C_{total} = 0.48 \ \$, T_{total} = 8 \ \text{hours} \tag{10}$$

### A.8.3 COST AND TIME ESTIMATION FOR HUMAN ANNOTATION

The average annotation time per image is assumed to be 15 seconds. Thus, the total annotation time for one annotator is:

$$T_{total}^{human} = 80,000 \times 15 = 1,200,000 \ \text{sec} = 333 \ \text{hours} \tag{11}$$

Considering full-time jobs with 8 working hours per day, the annotation process needs more than 40 days for one annotator.

The cost of human annotation varies by region, we take 13 $ per hour as a rough estimation. The total cost for human annotation is thus:

$$C_{human} = 333 \times 13 = 4,000 \ \$ \tag{12}$$

### A.9 ALGORITHM

In this section, we present the full and detailed algorithm of our online ICD pipeline (see Algorithm. 2) with the definition of the two functions SELECTDEMO (see Algorithm. 3) and TTS (see Algorithm. 4).

---

**Algorithm 2** Online In-Context Distillation

---

**Input:** Dataset $\mathcal{D}$, Student $S$, Teacher $T$, Pool $\mathcal{P}$, Multimodal Encoder $E$, Threshold $\delta$
**Output:** Prediction $\hat{A}$ for each samples $x$ in $\mathcal{D}$

1:  **for** each $x = (I, Q) \in \mathcal{D}$ **do**
2:      Predict with Zero-shot and estimate uncertainty $\hat{A}, u \leftarrow S(x)$
3:      ▷ *Uncertainty check, see Sec. 4.4*
4:      **if** $u < \delta$ **then**
5:          **return** $\hat{A}$
6:      **else**
7:          ▷ *Demonstration selection, see Sec. 4.3*
8:          Select the demonstration set $\mathcal{G} \leftarrow \text{SELECTDEMO}(x, \mathcal{P}, E)$
9:          Predict with ICL and estimate uncertainty $\hat{A}', u' \leftarrow S(x \mid \mathcal{G})$
10:         **if** $u' < u$ **then**
11:             Update the prediction $\hat{A} \leftarrow \hat{A}'$
12:         **end if**
13:         **if** $u' \geq \delta$ **then**
14:             ▷ *Refine with TTS, see Sec. 4.2*
15:             Use teacher model to annotate and refine $A' \leftarrow \text{TTS}(T(x))$
16:             Concatenate the annotated sample to the pool $\mathcal{P} \leftarrow \mathcal{P} \cup \{(x, A')\}$
17:         **end if**
18:         **return** $\hat{A}$
19:     **end if**
20: **end for**

---

**Algorithm 3** SELECTDEMO$(x, \mathcal{P}, E)$

---

**Input:** Sample $x$, Pool $\mathcal{P}$, Encoder $E$, Selection Number $K_{ii}, K_{tt}, K_{it}$
**Output:** Selected demonstration samples

1:  ▷ *Extract features*
2:  $(F_i, F_t) \leftarrow E(\mathcal{P})$
3:  $(\hat{f}_i, \hat{f}_t) \leftarrow E(\hat{x})$
4:  ▷ *Rank by cross-modal similarity*
5:  Selecting top K from the Index $\leftarrow \text{TopK}(CosineSimilarity(F_t, \hat{f}_i), K_{it})$
6:  Filter features: $F_i, F_t \leftarrow F_i[\text{Index}], F_t[\text{Index}]$
7:  ▷ *Rank by image-image similarity*
8:  Index $\leftarrow \text{TopK}(CosineSimilarity(F_i, \hat{f}_i), K_{ii})$
9:  Filter features: $F_i, F_t \leftarrow F_i[\text{Index}], F_t[\text{Index}]$
10: ▷ *Rank by text-text similarity*
11: Index $\leftarrow \text{TopK}(CosineSimilarity(F_t, \hat{f}_t), K_{tt})$
12: Filter features: $F_i, F_t \leftarrow F_i[\text{Index}], F_t[\text{Index}]$
13: **return** $\mathcal{P}[\text{Index}]$

---

**Algorithm 4** TTS$(T, x)$

---

**Input:** Teacher model $T$, Data sample $x$, Iteration K
**Output:** Annotation $\hat{A}$

1:  $\mathcal{A} \leftarrow \emptyset$
2:  ▷ *Best-of-N with multiple inferences*
3:  **for** $k = 1$ to $K$ **do**
4:      $A_k \leftarrow T(x)$
5:      $\mathcal{A} \leftarrow \mathcal{A} \cup \{A_k\}$
6:  **end for**
7:  ▷ *Keep only consistent prediction*
8:  **if** Consistent prediction: $\forall i \neq j, A_i, A_j \in \mathcal{A}, A_i = A_j$ **then**
9:      **return** $A_0$
10: **end if**

---

Table 7: Evaluation of ICL classification tasks. Our ICD consistently improves over zero-shot performance. Baseline methods are tested with zero-shot and up to 5 demonstrations for ICL. The performance of fine-tuned models and GPT-4o is included as a reference. The best results (excluding oracle) of each model are highlighted in bold. Oracle labels represent the upper bound with human annotations.

| Baseline | Method | GTSRB Zero shot | 1 shot | 2 shot | 3 shot | 5 shot | WikiArt Zero shot | 1 shot | 2 shot | 3 shot | 5 shot | CUB Zero shot | 1 shot | 2 shot | 3 shot | 5 shot |
|---|---|---|---|---|---|---|---|---|---|---|---|---|---|---|---|---|
| GPT-4o | - | 68.1 | | - | | | 56.3 | | - | | | 76.4 | | - | | |
| InternVL2.5 | Fine-tuning | 47.2 | \multicolumn With oracle labels: 86.7 / With teacher labels: 79.4 | | | | 34.1 | With oracle labels: 60.7 / With teacher labels: 58.7 | | | | 10.5 | With oracle labels: 84.8 / With teacher labels: 83.5 | | | |
| MMICL | Oracle | | 50.8 | 77.4 | 79.3 | 79.0 | | 34.3 | 42.3 | 44.2 | 36.2 | | 40.6 | 68.7 | 71.6 | 72.6 |
| | Self-labeling | 24.1 | 23.4 | 27.2 | 27.5 | 25.1 | 32.8 | 30.5 | 32.3 | 32.9 | 28.7 | 0.9 | 7.0 | 7.5 | 7.6 | 7.8 |
| | ICD | | 46.5 | 65.6 | **68.2** | 67.3 | | 33.4 | 41.9 | **43.0** | 34.3 | | 38.0 | 62.4 | **65.4** | 64.5 |
| LLaVA-1.5 | Oracle | | 85.2 | 85.4 | 85.3 | 84.3 | | 49.4 | 47.4 | 48.5 | 47.3 | | 65.0 | 74.9 | 79.3 | - |
| | Self-labeling | 2.6 | 2.0 | 2.3 | 2.2 | 2.1 | 16.4 | 17.3 | 17.0 | 18.3 | 17.8 | 2.5 | 3.0 | 2.6 | 2.7 | - |
| | ICD | | 70.6 | 72.0 | 72.5 | **72.7** | | 49.5 | **49.9** | 49.7 | 47.2 | | 57.8 | 67.6 | **70.6** | - |
| InternVL2.5 | Oracle | | 57.3 | 69.6 | 78.8 | 80.1 | | 42.4 | 41.3 | 44.7 | 43.0 | | 54.9 | 59.7 | 62.7 | 61.2 |
| | Self-labeling | 47.2 | 44.6 | 46.1 | 47.7 | 48.3 | 34.1 | 33.2 | 32.6 | 32.5 | 30.4 | 10.5 | 10.5 | 10.7 | 11.8 | 11.3 |
| | ICD | | 64.9 | 63.5 | 66.1 | **66.6** | | 41.3 | 43.7 | **44.1** | 42.7 | | 54.6 | 58.3 | **62.9** | 57.2 |
| LLaVA-OneVision | Oracle | | 85.5 | 85.9 | 85.6 | 86.1 | | 54.2 | 54.2 | 55.8 | 54.7 | | 80.0 | 79.4 | 80.7 | 80.9 |
| | Self-labeling | 42.6 | 42.2 | 41.9 | 41.6 | 41.8 | 43.1 | 40.3 | 40.0 | 40.4 | 38.9 | 26.9 | 27.6 | 27.1 | 27.2 | 26.5 |
| | ICD | | 72.0 | 72.5 | 72.6 | **74.3** | | 52.0 | 51.8 | **53.3** | 52.7 | | 72.4 | **73.1** | 73.1 | 72.9 |

Table 8: Evaluation of ICL generative tasks. Our ICD effectively enhances recent small VLMs. Baseline methods are tested with zero-shot and up to 5 demonstrations for ICL. The performance of fine-tuned models and GPT-4o is included as a reference. The best results (excluding oracle) of each model are highlighted in bold. Oracle labels represent the upper bound with human annotations.

| Baseline | Method | Flickr30k Zero shot | 1 shot | 2 shot | 3 shot | 5 shot | PMC-VQA Zero shot | 1 shot | 2 shot | 3 shot | 5 shot | VQA-AD Zero shot | 1 shot | 2 shot | 3 shot | 5 shot |
|---|---|---|---|---|---|---|---|---|---|---|---|---|---|---|---|---|
| GPT-4o | - | 24.6 | | - | | | 55.8 | | - | | | 41.0 | | - | | |
| InternVL2.5 | Fine-tuning | 10.7 | With oracle labels: 16.9 / With teacher labels: 17.3 | | | | 34.5 | With oracle labels: 54.1 / With teacher labels: 52.8 | | | | 36.0 | With oracle labels: 60.0 / With teacher labels: 54.0 | | | |
| MMICL | Oracle | | 28.6 | 29.0 | 28.5 | 27.7 | | 35.9 | 36.0 | 34.3 | 34..4 | | 20.0 | 28.0 | 29.0 | 32.0 |
| | Self-labeling | 24.3 | 23.5 | 22.4 | 21.5 | 19.0 | 34.7 | 34.2 | 33.2 | 32.5 | 32.3 | 22.0 | 23.0 | 27.0 | 27.0 | 31.0 |
| | ICD | | **28.9** | 28.8 | 27.7 | 27.3 | | 34.6 | **35.4** | 31.7 | 33.2 | | 24.0 | 32.0 | 31.0 | **35.0** |
| LLaVA-1.5 | Oracle | | 20.9 | 23.0 | 23.1 | 23.6 | | 21.8 | 20.3 | 18.8 | 20.8 | | 17.0 | 18.0 | 19.0 | 20.0 |
| | Self-labeling | 25.1 | 18.5 | 20.4 | 21.4 | 21.0 | 19.1 | 30.0 | 33.3 | **34.9** | 31.8 | 32.0 | 18.0 | 19.0 | 17.0 | 24.0 |
| | ICD | | 21.3 | 22.4 | 22.2 | **22.5** | | 21.6 | 20.3 | 18.6 | 21.2 | | **28.0** | 24.0 | **28.0** | 19.0 |
| InternVL2.5 | Oracle | | 12.7 | 12.7 | 12.1 | 11.7 | | 42.1 | 39.1 | 39.2 | 38.2 | | 38.0 | 40.0 | 43.0 | 37.0 |
| | Self-labeling | 10.7 | 11.5 | 11.8 | 11.2 | 11.6 | 34.5 | 37.5 | 36.4 | 35.1 | 34.9 | 19.0 | 10.0 | 19.0 | 21.0 | 21.0 |
| | ICD | | 13.6 | 13.7 | **14.1** | 13.5 | | **40.5** | 39.9 | 38.9 | 35.7 | | 31.0 | 35.0 | **37.0** | 33.0 |
| LLaVA-OneVision | Oracle | | 23.1 | 26.5 | 27.7 | 28.1 | | 48.2 | 49.1 | 48.3 | 47.5 | | 47.0 | 51.0 | 58.0 | 56.0 |
| | Self-labeling | 26.7 | 20.7 | 24.0 | 24.7 | 26.0 | 47.5 | 45.6 | 48.2 | 48.0 | 47.2 | 37.7 | 41.0 | 39.0 | 41.0 | 43.0 |
| | ICD | | 22.8 | 26.1 | 26.9 | **28.2** | | 48.0 | **48.7** | 48.5 | 47.7 | | 49.0 | 49.0 | **50.0** | 48.0 |

## A.10 Multi-Shot ICL Experiments

In this section, we present the experiments with the offline-gathered pool, where the teacher annotates the entire unlabeled set. With the labeled pool, we perform ICL with different numbers of demonstrations to show the benefit of providing more demonstrations and to compare the performance with oracle labels. The results are presented in Tab. 7 and Tab. 8. Our experiments confirm that 3-shot ICL is the most effective choice in general and with more demonstrations (i.e., 5 shots), the performance instead degrades. We attribute this phenomenon to the extended input length with more demonstrations and the potential noise in the demonstrations.

## A.11 Discussion of Datasets

### A.11.1 Description of Datasets for Evaluation

Following our analysis on Sec. 3, we consider classification-based tasks to show a good potential of ICL with necessary capabilities fulfilled, i.e. image-text alignments. Furthermore, we consider the more challenging generative tasks, such as VQA and image captioning, under the condition that they designate specific domain knowledge. For classification tasks, GTSRB Stallkamp et al. (2012)

contains over 50,000 traffic sign images across 43 classes for recognition tasks. WikiArt Tan et al. (2019) includes paintings from 27 artistic styles with more than 80k images. CUB Wah et al. (2011) comprises 11,788 bird images from 200 species for fine-grained classification. For generative tasks, PMC-VQA Zhang et al. (2023a) is a biomedical VQA dataset that contains a total of 227k VQA pairs of 149k images. VQA-AD Atakishiyev et al. (2023) focuses on synthetic autonomous driving scenes, covering 5 driving actions with 250 frames for training and 100 frames for testing. Flickr30k Young et al. (2014) features 31,000 images with human-annotated captions for image-text modeling.

### A.11.2 EVALUATION ON GENERIC DATASETS

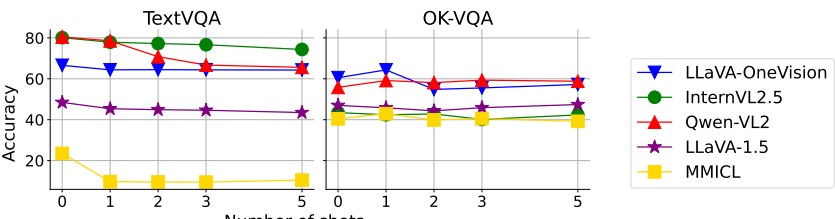

Figure 14: ICL experiments on generic datasets. Most models cannot improve with demonstrations.

Based on our analysis in Sec. 3, we conclude that ICL favors specific tasks over generic tasks. To further justify our finding, we show the ICL experiments on two representative generic VQA datasets: TextVQA Singh et al. (2019) and OK-VQA Marino et al. (2019). TextVQA comprises 45,336 questions with 28,408 images, requiring models to read and reason about text within images. In contrast, OK-VQA includes over 14,000 questions with 10 diverse categories, necessitating external knowledge or commonsense reasoning to provide correct answers. In these experiments, we use the oracle label for the demonstration. The results are shown in Fig. 14. We observe that all models tested struggle to improve with demonstrations, which supports our remark in Sec. 3. As the dataset is generic and diverse, it is more challenging to find semantically meaningful demonstrations that would help the model better understand the task. We leave further investigation of ICL on these challenging datasets as future work.

### A.12 DIFFERENT SIZE OF UNLABELED SET FOR ICD

In the main experiments, we consider annotating all the training samples to ensure the diversity and relevance of the demonstrations. In practice, there is no need to annotate the entire training set, and one can consider applying our framework whenever a few samples are available from a new domain or task. As shown in Fig. 15, the final performance remains close when we reduce the annotation size from the full set to 25% of the data. However, there is a clear gap between 5% and the full set, which confirms the necessity to have diverse demonstrations for ICL. Moreover, we show in the figure the performance of our online ICD, where the percentage of the gathered pool is automatically collected with the uncertainty-aware selection. We also report the online ICD results in the same figure, with respect to the reduced annotation rate $T(x)$. It is interesting to see that our online ICD achieves significantly higher accuracy with fewer data annotated, which confirms the effectiveness of an active and selective annotation strategy.

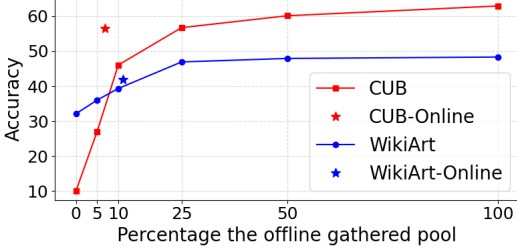

Figure 15: Comparison of different annotation scales in our framework.

### A.13 JUSTIFICATION OF THE UNCERTAINTY MEASURE

In this section, we justify our choice of uncertainty measure, i.e., average entropy of the generated sequence (see Sec. 4.4 and Eq. 3). First, we show the negative correlation between the uncertainty score and the prediction accuracy. Moreover, we compare different uncertainty estimation approaches.

#### A.13.1 CORRELATION BETWEEN UNCERTAINTY AND PREDICTION ACCURACY

We conduct a statistical analysis to quantify the correlation between these two terms. Specifically, we estimate the Point Biserial Correlation Kornbrot (2014), which is dedicated to measuring the correlation between a continuous variable, i.e., uncertainty score, and a binary variable, i.e., prediction accuracy. The correlation score ranges from $-1$ to $1$, where a value closer to $-1$ means a stronger negative correlation between the two variables. We first perform zero-shot predictions on the test data and estimate the uncertainty as described in Sec. 4.4, and we repeat the experiments on CUB and WikiArt datasets with the LLaVA-OneVision model. We obtain $-0.400$ with a p-value $4.82 \times e^{-96}$ on CUB and $-0.359$ with a p-value of $9.73 \times e^{-38}$ on WikiArt, which indicates a statistically significant negative relationship between uncertainty and prediction accuracy. This suggests that the model tends to be more accurate when it is confident, and less accurate when it expresses higher uncertainty. This test justifies the effectiveness of this simple approach as an indicator of the prediction reliability.

#### A.13.2 COMPARISON OF DIFFERENT ENTROPY-BASED UNCERTAINTY

We further justify our choice by comparing different strategies for calculating the entropy. In our Eq. 3, we estimate the uncertainty as the average entropy across all indices. We denote our strategy as *All sequence* as it considers the entire sequence. However, there are other alternatives. For instance, one can consider the probability distribution of only the first token or the last token (i.e., the famous <EOS> token) instead of all sequence to calculate the entropy. We show the comparison in Tab. 9 with Point Biserial Correlation. We observe that the probability distribution of the

Table 9: Comparison of different strategies to estimate entropy-based uncertainty. Our chosen method is among the most significant ones.

| Dataset | Method | Correlation | P-Value |
|---------|--------|-------------|---------|
| CUB | EOS | $-0.013$ | $0.537$ |
| | First token | $-0.409$ | $3.02 \times e^{-96}$ |
| | All sequence | $-0.400$ | $4.82 \times e^{-96}$ |
| WikiArt | EOS | $-0.120$ | $3.09 \times e^{-5}$ |
| | First token | $-0.358$ | $1.67 \times e^{-37}$ |
| | All sequence | $-0.359$ | $9.73 \times e^{-38}$ |

<EOS> token is not always statistically significant. Instead, considering only the first token or the entire sequence leads to significant correlation between the uncertainty and prediction accuracy. These results confirm our design choice for the uncertainty estimation.

#### A.13.3 COMPARISON OF DIFFERENT UNCERTAINTY MEASURES

We also considered other uncertainty measures in our experiments. According to recent studies Yang et al. (2023); Heo et al. (2024), off-the-shelf methods for uncertainty estimation can be categorized into 3 groups: **entropy-based**, **verbalized**, and **multiple-inference-based**. We first compare entropy-based and verbalized uncertainty, as both of them can be obtained with a single inference. This is an important advantage for our online ICD as it requires the student model to give an instant response

Table 10: Comparison of different strategies to estimate uncertainty. Our chosen method is statistically significant while the other one is not.

| Dataset | Method | Correlation | P-Value |
|---------|--------|-------------|---------|
| CUB | Verbalized | $0.13$ | $0.02$ |
| | Entropy | $-0.400$ | $4.82 \times e^{-96}$ |
| WikiArt | Verbalized | $-0.065$ | $0.259$ |
| | Entropy | $-0.359$ | $9.73 \times e^{-38}$ |

in the online evaluation. In this regard, uncertainty estimation with multiple inferences that lead to latency and computational overhead is not ideal for our scenario. The comparison is in Tab. 10. The system prompt for the model for verbalized uncertainty is *Estimate your uncertainty with a float value between 0 and 1, with 0 as no uncertainty and 1 as the maximal uncertainty.* We found that, unlike entropy-based uncertainty, verbalized uncertainty fails to reflect the reliability of the prediction.

Moreover, we compare with multi-inference-based uncertainty estimation. Specifically, we convert multi-inference uncertainty into a binary code, where consistent outputs are denoted as 0 for no uncertainty, and inconsistent outputs are denoted as 1. Similarly, we measure the Point Biserial Correlation between the entropy-based uncertainty and the uncertainty code from multiple inferences. We obtain $0.482$ with a p-value of $1.99 \times e^{-30}$ on WikiArt and $0.426$ with a p-value of $1.86 \times e^{-23}$ on CUB. This justifies that these two uncertainty measures are to some extent equivalent to each other. However, entropy-based uncertainty prevents the need for multiple inferences, which is thus a reasonable choice for our uncertainty estimation.

### A.14 LIMITATIONS AND BROADER IMPACTS

Our ICD framework assumes the model possesses sufficient ICL capability. When this assumption does not hold, the model may not benefit from our framework. While our goal is to promote the use of small VLMs in practical applications—where smaller models offer better efficiency and accessibility—we are currently constrained by the capabilities of existing models. In particular, our framework does not extend to tiny VLMs with fewer than 4B parameters, which lack adequate ICL ability. We anticipate that ongoing progress in VLM research will yield more capable small models, enabling our approach to generalize to even smaller scales and more realistic deployment scenarios. Moreover, while we focus on a training-free paradigm where we rely on test-time computation to enhance the performance, the framework can be extended to incorporate in-context training for a model as a post-pretraining step to improve the performance at inference time.

Regarding the broader impact of our method, by avoiding full model fine-tuning and reducing reliance on large teacher models, our ICD lowers the energy footprint typically associated with adapting VLMs to new tasks. This makes it more accessible and sustainable for users with limited computational resources. Additionally, by facilitating the practical deployment of compact models, our approach may support broader adoption of multimodal AI in low-resource settings, such as education, environmental monitoring, or assistive technologies. At the same time, we acknowledge that our method builds on existing pretrained models and may inherit their limitations, including potential biases. Responsible use and continued evaluation in downstream applications remain important considerations.

### A.15 JUSTIFICATION OF OUR SELECTION MECHANISM

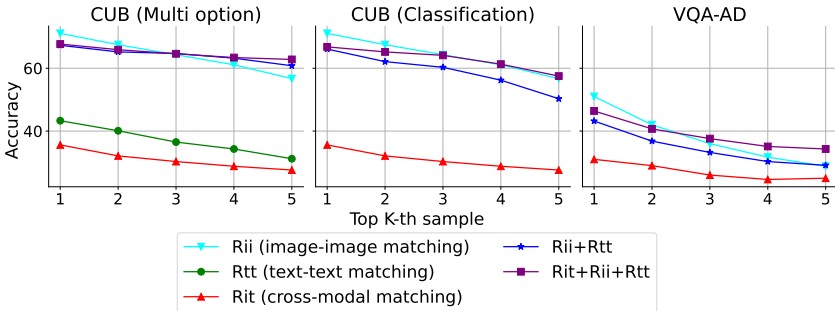

Figure 16: Comparison of the demonstration accuracy. Our cross-modal matching helps ensure a reasonable demonstration selection when the query text is generic. The text-text selection accuracy is omitted for clarity of the figure on CUB (Classification) and VQA-AD because of random selection.

We further justify our selection mechanism. We define the demonstration accuracy for a classification task as the proportion of demonstrations that belong to the same class as the query sample. We report this value as it is a direct measure of the quality of the demonstrations and the quality of the selection, and it is positively correlated with a better ICL performance. Moreover, to better illustrate the issue of generic query text, we utilize the CUB dataset with two question formats. In the standard classification task, the question is generic, i.e., *What species is this bird?*. In contrast, once converting the CUB dataset for multi-option VQA, the question contains additional information from the options, e.g., *What species is this bird? A. California Gull B. American Pipit*. In the former, since the question's embedding contains no information about the class, the text-text matching is ineffective. Nevertheless, in the latter, the options provide hints for the candidate classes and can be used for text-text matching. VQA-AD is included as it also contains a generic query question.

Therefore, we apply different strategies that we present in Sec. 4 to select demonstrations and report the demonstration accuracy in Fig. 16. Image-image matching $R_{ii}$, text-text matching $R_{tt}$, and cross-modal matching $R_{it}$ are denoted as Rii, Rtt, Rit, respectively, in the figure. Note that we omit the text-text matching in the CUB (Classification) and VQA-AD setting to improve the visibility, as the matching with a generic query question is a random selection. First, we found that image-image matching is the most effective selection mechanism in most cases, especially for a task that is based on image classification. Moreover, questions with options enable effective text-text selection. Next, the cross-modal matching is not a strong selection mechanism as it relies on cross-modal matching capability of the multi-modal encoder. This explains why we only use it as a pre-selection mechanism to filter out potential noise. However, the cross-modal matching significantly enhances the selection when the text-text matching is not effective, as it helps eliminate irrelevant samples and reduces the noise. Lastly, we observe that the pre-selection of our cross-modal matching enhances the selection accuracy of the fourth or fifth demonstration, as it is useful in eliminating the noise from the candidate samples. We further show the comparison on VQA-AD, which contains more complex scenes in each image than classification tasks, leading to a degraded reliability on image-image only selection. In this case, the improvement of our proposed cross-modal matching is more noticeable. This justifies our design choice and highlights the importance of having an effective selection mechanism.

### A.16   LoRA fine-tuning details

We compare our ICD with LoRA-based fine-tuning. In principle, parameter-efficient tuning methods are significantly less compute-intense than full-parameter tuning. However, we still observe an advantage of our ICD over this method, showing its best performance with minimal compute budget.

For LoRA fine-tuning, we follow the implementation of the official codebase of InternVL2.5. Specifically, we use a rank of $r = 16$. The LoRA adapter is inserted to the language model of the VLMs. In our resource constraint setting, to avoid a substaintial computational overhead, we apply an online learning paradigm with LoRA. Specifically, we wait until the teacher model annotate 64 samples to form a mini-batch. With this mini-batch, we conduct one gradient descent. After each step, we evaluate the test data with the updated LoRA adapter and the frozen backbone to calculate the accumulated accuracy. The associated computational cost consists of the training cost and the inference cost.

