# OpenReview forum: "Online In-Context Distillation for Low-Resource Vision Language Models"
_ICLR.cc/2026/Conference — Submitted to ICLR 2026_

### Official Review · Reviewer_r3Rf · 2025-10-15

**Soundness:** 3
**Presentation:** 3
**Contribution:** 3
**Rating:** 4
**Confidence:** 4

**Summary:**

The paper proposes Online In-Context Distillation (ICD): a training-free framework where a small VLM (student) improves at test time by retrieving sparse demonstrations labeled by a stronger teacher VLM, then performing in-context learning (ICL) with those demos. Key modules include: (i) teacher test-time scaling (Best-of-N) to reduce noisy labels, (ii) cross-modal demonstration selection (image↔text) to pick useful demos, and (iii) uncertainty-aware querying to limit teacher calls. Experiments across classification, VQA, and captioning show sizable gains for small VLMs (e.g., LLaVA-OneVision 7B from 42.6% → 70.8%) with low teacher query rates (~14.7%) and favorable compute/cost trade-offs versus fine-tuning.

**Strengths:**

Tackles a real-world, practical problem. The paper focuses on improving small vision-language models (VLMs) without training, which is super relevant for resource-constrained settings. I appreciate that it doesn’t just chase benchmarks — it thinks about practical deployment constraints like compute and memory.

Nice and well-thought-out design. The proposed framework makes a lot of sense: use a strong teacher only when the student is unsure, and make that decision based on uncertainty. Also, I like the three-step demo selection process — it’s simple but smart.

Surprisingly strong results given how little supervision is used. It’s impressive that the student model gets big accuracy gains while only querying the teacher around 15–18% of the time. Sometimes it even outperforms the teacher’s own zero-shot predictions.

Cares about cost and efficiency. The paper goes beyond accuracy and includes cost/CO₂ comparisons versus online LoRA fine-tuning. This is rare but really helpful for practitioners.

Clear scope and responsible framing. The authors are upfront about where in-context learning (ICL) helps and where it doesn’t. They also explain when fine-tuning might still be necessary, which is great for clarity.

**Weaknesses:**

Teacher cost and quality could use more analysis. The paper introduces a Best-of-N strategy to clean up noisy labels from the teacher, which is nice. But we don’t really get a deep dive into how label quality scales with N, or how sensitive results are to teacher sampling strategies. Also, it would be good to know how much this costs across more datasets.

Statistical reporting is a bit light. Some of the results (especially in Table 1) look strong, but we don’t get confidence intervals or variance over seeds/streams. That makes it harder to tell whether gains are truly consistent. Not enough ablation on key design choices. The demo selection pipeline has several moving parts (e.g., cross-modal stages, entropy threshold δ, pool size, number of shots), but we only see limited ablation on these, especially in the actual online scenario.

Benchmark coverage could be broader. It’s good that the authors test across classification, VQA, and captioning — but I would’ve liked to see more diverse real-world tasks (like TextVQA or OK-VQA) included in the main results, rather than tucked into the appendix. Latency is underreported. The whole idea is to do smart inference in real time, but there’s no mention of the actual latency per student pass, with/without ICL, and with teacher queries. That’s a key part of making this deployable.

**Questions:**

How do you pick δ (uncertainty threshold)? Is it task-specific or fixed across benchmarks? Did you try adaptive schemes? How sensitive are your results to the choice of δ? Can you explain the demo selector details a bit more? What values do you use for Kₜₜ, Kᵢₜ, etc.? And what similarity functions are you using in the three stages? How reliable are teacher labels under Best-of-N? Do you have stats like agreement rate or correctness vs. N? That would help quantify how noisy the teacher is.

What happens if your demo pool gets corrupted? Any chance bad examples get added and mislead the student? Do you have any strategy for removing low-quality demos? Can you report actual latency? Say I run this on an edge device with remote teacher access — what’s the real-world latency per sample?

Could you add more statistical reporting? Mean ± std over multiple random streams or seeds (especially for Table 1) would be helpful.

Did you try other ways to improve teacher label quality?Like reranking with self-consistency or temperature scaling? Would love to hear about alternatives to Best-of-N. Will you release code, prompt templates, and configs? That would really help with reproducibility — especially since the method depends on several pipeline pieces.

If you’re able to address the issues around uncertainty thresholding, demo selection ablations, and latency, I’d be happy to consider bumping up my score!

---

### Official Review · Reviewer_cF6c · 2025-10-30

**Soundness:** 4
**Presentation:** 4
**Contribution:** 2
**Rating:** 6
**Confidence:** 3

**Summary:**

The paper presents a study on how different VLMs can improve provided additional context from a pool, according to their size. The authors identify a scale of models that are prone to benefitting from such additional context, and propose a method for automatic selection based on teacher models rather than on human annotations. Based on the confidence of the model's predictions, a three step algorithm is devised to provide the student with additional context where necessary. The results of the study and the proposed approach show that the model incurs in much less computational demand than existing methods.

**Strengths:**

The presented scale-based study is convincing and well defined. A good variety of models is studied, and a good set of varying complexity models is included. A proper carbon-emission informed study is presented, motivating their proposed approach.

The paper is well written, presented, and threaded. It is a good piece of work with several takeaway messages and a proper breakdown of the different studies included.

**Weaknesses:**

The technical contribution is in my opinion a bit limited. Under the hood, the authors propose to use GPT-4o to replace human annotators to provide the context in an on-demand manner.

Referring to the method as distillation is a bit misleading in my opinion. Basically what the authors are doing is replacing the answer for the pool of < query, image >, typically hand annotated, by the output of a GPT-4o. Such technique is closer to pseudolabelling than to distillation.

The paper appeals to the save in computational cost of the proposed in-context distillation. The main idea is that the small VLM will retrieve a pool of Q,I, which will be sent to the server where GPT-4o is running to provide the answers. My question here is, is this retrieval applied in a per-query basis or is it done once before the use of the small VLM? In other words, it is not clear to me (please clarify if I missed this in the paper) if there is a need for a pool for every potential query asked to the small VLM. In that case the computational benefit is a bit unclear to me, and it would call for a baseline where the answer is directly given by GPT-4o; i.e. a cost/accuracy measurement of GPT-4o for each dataset.

Overall, the paper has  interesting findings, and its quality and presentation are borderline accept to me. The main limitation is that the paper's technical contribution is rather stretched, putting under question whether it should be accepted at a top venue such as ICLR.

**Questions:**

One of my main concerns refers to the pool selection, i.e. whether this is retrieved in a per-query basis or globally for a given VLM. I would like to have a clear idea of whether one needs access to a server to use the small VLM to perform in-context retrieval. Please clarify this aspect in the rebuttal.

---

### Official Review · Reviewer_2w8L · 2025-11-01

**Soundness:** 2
**Presentation:** 3
**Contribution:** 2
**Rating:** 2
**Confidence:** 4

**Summary:**

The work explores the effectiveness of ICD techniques for 4-12B sizes VLMs. To save on labeling cost, a larger teacher generates example labels at test time. In order to construct the method, an analysis is  conducted to identify suitable choices. The performance is enhanced using a  cross-modal demonstration selection strategy, a teacher test-time scaling to reduce noise, and student uncertainty measurements.

**Strengths:**

- Overall, the paper is easy to follow
- Good results compared with the baselines implemented
- The direction of using a larger (V)LLM to provide annotations is sound

**Weaknesses:**

- Novelty wise, generating labels with a bigger teacher model is a standard technique, in fact many of the standard VLLM datasets are constructed as such (e.g. ShareGPT-4V, parts of LLaVA data itself). In fact, the initial version of LlaVA conversational data was generated using a GPT model without images (based on image content description). Similarly, distillation from bigger teachers was used with success for various VLMs, eg.: LLaVA-KD: A Framework of Distilling Multimodal Large Language Models, Cai et al, 2025;  VLsI: Verbalized Layers-to-Interactions from Large to Small Vision Language Models, Lee et al 2025 etc.
Given this, I find that the contribution of using a bigger model to generate data carries insufficient novelty.

- The current comparison needs to be expanded to a full suite of benchmarks. See LlaVA-OV benchmarks for example. Many of the datasets the models are evaluated are zero-shot, so the setting described should remind valid.

- The work claims that the proposed approach is more efficient and suitable than finetuning. However, results shown in Table 8 suggest that the proposed approach saturates after 2-3 examples. In contrast, finetuning can scale in principle indefinitely. Moreover, only LoRA adaptation is considered. Ideally comparison with other finetuning strategies, under different learning rates should be considered too (e.g: prompt learning etc).

- Currently, the efficiency measure only the train/adaptation time. However, the real cost occurs at test time. Adding examples in context increases notably memory usage and cost. For a fair comparison, it will be good to report both train and test-time costs.

- The CO2 metric used while commendable is hard to interpret. What are the FLOPs count and wall clock time?

- Regarding baselines: [1] CoT is generally used for eliciting reasoning for solving tasks. Not all tasks would be suitable for this, hence are any of the tasks used suitable? Could the authors perhaps elaborate? Many of them are somewhat simple, as an example flickr consists of describing the images with very short captions, where really improving the performance when measure with BLEU score is more of adjusting the style of the caption, as this models can already caption images pretty well by they get the style wrong.

- I find the usage of small for classifying a 4-12B scale a bit misleading. My expectation was to have sub-4B as small, perhaps this could be made clear early one. The reason behind this is that 7B for example is just the "default" size for most of the VLMs, and in literature (see SmalVLM for example), small has different range. Moreover, in Line 073 Awadalla et al, 2023 is cited as an example of work with ICL for large VLMs, however the work in question studies model sizes between 3-9B. This results in inconsistent claims as it suggests that in fact other works studies the problem in the context of the present definition of small models.

**Questions:**

- How is the behavior different for LLM vs VLMs? It's not very clear to me from reading the Section. Given that similar studies were done for LLMs, the authors should highlight how the two problems differ.

- Does the method work with multi images?

- Given that the work aims to match the 0-zhot performance of the teacher, what is the cost of a) running the teacher vs b) adding the in-context examples as extra KV-cache during every subsequent inferences.

---

### Official Review · Reviewer_KwcU · 2025-11-01

**Soundness:** 3
**Presentation:** 3
**Contribution:** 3
**Rating:** 4
**Confidence:** 3

**Summary:**

This paper introduces Online In-Context Distillation (ICD), a training-free framework designed to enhance the
performance of small vision-language models (VLMs) in low-resource settings. The method leverages a large
teacher model to generate on-the-fly demonstrations, which are selectively retrieved and used by a small
student model during inference via in-context learning (ICL). The authors first establish that VLMs with at least
4B parameters are necessary for effective ICL and demonstrate the advantages of ICL over fine-tuning under
compute constraints. The proposed ICD framework incorporates three key components:

1. Test-time scaling (TTS) to improve teacher annotation quality,

2. Cross-modal demonstration retrieval (image-text, image-image, text-text) for robust selection,

3. Uncertainty-aware conditioning to minimize teacher queries.

Extensive experiments across classification, VQA, and captioning tasks show that ICD significantly boosts
student performance (e.g., +14.8% on average) with minimal annotation overhead (∼14.7% teacher query
rate).

**Strengths:**

- The paper provides a rigorous experimental quantification of the parameter scale (≥4B) required for
effective in-context learning in Vision-Language Models, alongside a detailed cost-benefit comparison
with fine-tuning. This offers crucial guidance for selecting and evaluating smaller models in resource-
constrained environments.
- By deconstructing the complex "operator induction" task into subtasks—"Image OCR → Operator
Reasoning → Result Calculation"—the work delivers profound diagnostic insights. It clearly pinpoints core
bottlenecks in multimodal understanding, specifically the disconnection between visual perception and
linguistic reasoning.
- The ICD framework integrates several components—Test-Time Scaling, cross-modal retrieval, and
uncertainty estimation—into a coherent, training-free system. Its plug-and-play nature makes it
exceptionally suitable for deployment on resource-limited edge devices.
- The study presents a thorough comparison of the practical costs (both time and financial) associated with
different annotation strategies, including human, self-, and teacher-based labeling. It robustly
demonstrates that ICD achieves an superior balance between performance gains and
annotation/computational overhead.

**Weaknesses:**

- The close alignment between the reported performance improvement (14.8%) and the teacher query
rate (14.7%) raises a critical question about the source of the gains. Moreover, the fact that the method
still fails to match GPT-4o on several tasks, despite this strategic querying, suggests a fundamental
limitation. It remains unclear whether the improvement stems from genuine enhancement of the
student's capabilities or simply from rerouting uncertain queries to the teacher. A direct experiment that
substitutes the teacher's answers for the student's on selected queries is necessary to isolate the effect
of mere answer forwarding.
- A key observation states that small VLMs significantly lag behind large teachers in multimodal ICL, even
when their pure-language performance is comparable. This conclusion is primarily drawn from
experiments with specific models like GPT-4o. Given the rapid evolution of large models, it remains an
open question whether this performance gap persists with newer, more powerful models (e.g., GPT-5,
Qwen3-VL), potentially limiting the long-term relevance of the finding.
- While the proposed three-stage retrieval strategy is effective, key ablation studies are missing. For
instance, how does the performance of a combination of only Rit and Rtt compare to the full three-step
method or a naive Rtt-only approach? This is crucial for justifying the necessity of each component.
- The selection strategy should be a core component of the method, yet there is a lack of ablation studies
on this part. For instance, the appendix has already shown that using image matching alone can achieve
strong performance. It would be informative to see the results if the selection strategy itself were based
solely on image matching.
- The use of "demonstration accuracy" (the proportion of demos from the same class as the query)
presupposes that "same-class" equates to "useful." However, for tasks requiring reasoning or diversity,
heterogeneous yet insightful demonstrations might be more beneficial. Furthermore, the analysis does
not address the potential negative impact of retrieving "same-class but incorrect" demonstrations, which
could introduce noise.
- The use of average token entropy as an uncertainty measure is well-validated for classification tasks via
its negative correlation with accuracy. However, for generative tasks like image captioning, where output
sequences are variable in length, entropy values are inherently sensitive to sequence length. The
reliability and effectiveness of this metric in such generative contexts therefore require further
demonstration.
- The paper presents results in a vacuum. To substantiate the claim of effectiveness, it is critical to ask:
How does the proposed method perform relative to other known models on these specific datasets?
Without benchmarking against a comprehensive set of existing results (even from non-distillation
approaches), the community cannot assess the actual contribution of this work. The authors must provide
these comparisons to demonstrate that their method offers a competitive advantage.

**Questions:**

1. To definitively rule out the explanation that the performance gains are primarily due to answer routing
rather than student learning, we strongly recommend a key ablation study: What would be the
performance if the system, upon selecting a query for the teacher, simply output the teacher's answer
and completely bypassed the student model? Comparing this baseline to the full method's performance
would directly quantify the contribution of the student's own processing and provide necessary evidence
for true distillation.

2. The conclusion regarding the significant ICL gap between small and large VLMs is drawn from
experiments with specific models (e.g., GPT-4o). How generalizable and long-lasting is this finding given
the rapid evolution of large models? Would the performance gap persist, or even diminish, when
evaluated against newer, more powerful models (e.g., future GPT-5 or Qwen3-VL)?

3. To justify the necessity of each component in the three-stage retrieval strategy, could the authors provide
ablation studies? For instance, how does the performance of using only Rit and Rtt compare to the full
three-step method or a naive Rtt-only approach?

4. Ablation studies are needed to identify the crucial components, in particular, the selection strategy.

5. The metric "demonstration accuracy" assumes that "same-class" demos are always "useful." However, for
complex reasoning tasks, could heterogeneous yet insightful demonstrations be more beneficial?
Furthermore, does the analysis account for the potential negative impact of retrieving "same-class but
incorrect" demonstrations?

6. The use of average token entropy is well-established for classification tasks. However, for generative
tasks like captioning where output length varies, how reliable is this metric? Could the authors further
demonstrate its effectiveness and justify its applicability in such generative contexts, given its inherent
sensitivity to sequence length?

7. How does the performance of the proposed method compare against a comprehensive set of
established baselines (including strong non-distillation models) on the exact same datasets? Without
such comparisons, it is impossible to determine whether the reported results represent a competitive
advancement or merely an improvement over a weak baseline.

---

### Meta-Review · Area_Chair_4WzK · 2025-12-30

**Summary:**

This paper received three negative scores (two marginally below the acceptance and one rejection), and one positive score with a marginally above the acceptance. Main concerns lie in the limited technical contribution (Reviewer cF6c and Reviewer 2w8L), missing the key ablation study (Reviewer KwcU), and the key analysis and explanation of pool selection(Reviewer cF6c). Moreover, the authors have not responded. After reading this carefully and considering the reviews, rebuttal, and the author's message, the Meta reviewer agrees with the concerns raised by the reviewers and recommends rejecting the paper.

**Reviewer Concerns:**

All concerns raised by the reviewers have not been addressed.

**Reviewer Scores:**

None

---

### Decision · Program_Chairs · 2026-01-26

Reject